

# Silicon cycle in the Tropical South Pacific: evidence for an active pico-sized siliceous plankton

Karine Leblanc[1], Véronique Cornet[1], Peggy Rimmelin-Maury[2], Olivier Grosso[1], Sandra Hélias-Nunige[1], Camille Brunet[1], Hervé Claustre[3], Joséphine Ras[3], Nathalie Leblond[3], Bernard Quéguiner[1]

[1]Aix-Marseille Univ., Université de Toulon, CNRS, IRD, MIO, UM110, Marseille, F-13288, France
[2]UMR 6539 LEMAR and UMS OSU IUEM - UBO, Université Européenne de Bretagne, Brest, France
[3]UPMC Univ Paris 06, UMR 7093, LOV, 06230 Villefranche-sur-mer, France

*Correspondence to*: Karine Leblanc (karine.leblanc@univ-amu.fr)

## 1 Abstract

This article presents data regarding the Si biogeochemical cycle during two oceanographic cruises conducted in the Southern Tropical Pacific (BIOSOPE and OUTPACE cruises) in 2005 and 2015. It involves the first Si stock measurements in this understudied region, encompassing various oceanic systems from New Caledonia to the Chilean upwelling between 8 and 34° S. Some of the lowest levels of biogenic silica standing stocks ever measured were found in this area, notably in the Southern Pacific Gyre, where Chlorophyll a concentrations are most depleted worldwide. Integrated biogenic silica stocks are as low as $1.08 \pm 0.95$ mmol m$^{-2}$, and are the lowest stocks measured in the Southern Pacific. Size-fractionated biogenic silica concentrations revealed a non-negligible contribution of the pico-sized fraction (<2-3 µm) to biogenic silica standing stocks, representing $26 \pm 12$ % of total biogenic silica during the OUTPACE cruise and $11 \pm 9$ % during the BIOSOPE cruise. These results indicate significant accumulation in this size-class, which was undocumented for in 2005, but has since then been related to Si uptake by *Synechococcus* cells. Our Si kinetic uptake experiments carried out during BIOSOPE confirmed biological Si uptake by this size-fraction. We further present diatoms community structure associated with the stock measurements for a global overview of the Si cycle in the Southern Tropical Pacific.



## 2 Introduction

Siliceous phytoplankton, especially diatoms, are often associated with nutrient-rich eutrophic ecosystems. However, the global budget of biogenic silica production by Nelson *et al.* (1995) already pointed out the importance of these organisms in oligotrophic areas where, despite their low concentration and due to the geographical extension of these systems, their silica production would be comparable to the total for all areas of diatomaceous sediment accumulation combined. However, studies that have documented the Si cycle in the Pacific Ocean, the largest oligotrophic area of the World Ocean, mainly focused on the Equatorial region, and the northern Subtropical gyre. This article presents the first set of field results from the Southern Pacific Ocean between 8 and 34° S spanning from New Caledonia over to the Chilean upwelling, and notably, from the most Chl*a*-depleted region at a worldwide scale (Ras et al., 2008): the South Pacific Gyre (SPG). Diatoms are known to contribute more importantly to primary production in meso- to eutrophic systems, yet several studies have emphasized that even if they are not dominant in oligotrophic regions, they may still contribute up to 10-20 % of C primary production in the Equatorial Pacific (Blain et al., 1997). In the oligotrophic Sargasso Sea, their contribution may be as high as 26-48 % of new annual primary production (Brzezinski and Nelson, 1995) and they may represent up to 30 % of Particulate Organic Carbon (POC) export (Nelson and Brzezinski, 1997). In the Eastern Equatorial Pacific (EEP), it has been shown that diatoms experience chronic Si-limitation along the Eastern Equatorial divergence in the so-called High Nutrient Low Silicate Low Chlorophyll (HNLSiLC) system (Dugdale and Wilkerson, 1998) as well as Si-Fe co-limitation (Blain et al., 1997; Leynaert et al., 2001). Furthermore, oligotrophic regions are known to experience considerable variability in nutrient injections leading to episodical blooms depending on the occurrence of internal waves (Wilson, 2011), meso-scale eddies (Krause et al., 2010) storms (Krause et al., 2009), or dust deposition events (Wilson, 2003). In nitrogen (N) depleted areas, punctual diatom blooms in the form of Diatom Diazotroph Associations (DDAs) are also known to occur and to contribute both to new primary production (Dore et al., 2008; Brzezinski et al., 2011) but also to benefit to non-diazotrophic diatoms through secondary N-release (Bonnet et al., 2016; Leblanc et al., 2016).

While biogenic silica was classically associated to the largest size fractions, especially microplankton, a series of recent studies have furthermore evidenced a role for picophytoplankton such as *Synechococcus* in the Si cycle, showing that this ubiquitous lineage is able to take up and





accumulate Si (Baines et al., 2012; Ohnemus et al., 2016; Krause et al., 2017; Brzezinski et al.,
2017). This was evidenced in the field in the Equatorial Pacific, the Sargasso Sea, as well as in
culture work, suggesting a widespread diffuse role for this organism, which could be more
prominent in oligotrophic environments where diatoms are in low abundance. In the EEP, and
despite very variable cellular Si content, *Synechococcus* represented for instance 40 % of water
column biogenic silica (BSi) inventory compared to diatoms in 2004, and twice that of diatoms the
following year (Baines et al., 2012). The role of small nano-sized diatoms has also probably been
overlooked and we recently pointed out their general occurrence at the worldwide scale and their
occasional regional importance in diatom blooms (Leblanc *et al.*, 2018).
Here we present the first set of field results from the Southern Pacific Ocean between 8 and 34° S
spanning from New Caledonia over to the Chilean upwelling, and notably, from the most depleted
Chl*a* region worldwide (Ras *et al.*, 2008), the South Pacific Gyre (SPG). Results were obtained
from two cruises carried out a decade apart following longitudinal sections first in the South Eastern
Pacific (SEP) between the Marquesas Islands and the Chilean upwelling, crossing the South Pacific
Gyre (BIOSOPE cruise, Oct-Dec 2004) and next in the Southern Western Pacific (SWP) between
New Caledonia and Tahiti (OUTPACE cruise, Feb-Apr. 2015). Very similar sampling strategies
and homogeneous analyses were conducted regarding the Si cycle and provide new data in this
under sampled region. We detail size-fractionated BSi inventories in the water column, Si export
fluxes, associated diatom community structure composition as well Si uptake and kinetic rates in
the Southern Pacific. Our key results show some of the lowest BSi stocks ever measured, which
may warrant for a revision of the contribution of oligotrophic areas to the global Si cycle, and
confirm recent findings of an active biological uptake of Si in the pico-sized fraction.
**3 Material and methods**
**3.1 Sampling strategy**
Results presented here encompass data from two French oceanographic cruises located in the
Southern Pacific Ocean (from 10 to 30° S), covering two transects with similar sampling strategies
of short and long duration stations. The BIOSOPE (BIogeochemistry and Optics SOuth Pacific
Experiment) cruise was undertaken in 2004, while the OUTPACE cruise took place in 2015, both
aboard the R/V *L'Atalante*. The BIOSOPE transect was sampled between the Marquesas Islands
(141° W, 8° S) and Concepción (Chile) (72° W, 35° S), between October 24[th] and November 12[th]





2004. The OUTPACE transect was sampled between New Caledonia (159° W, 22° S) and Tahiti
(160° W, 20° S) between February 18th and April 3rd 2015 (Fig. 1).

**3.2 Hydrology**

Water sampling and measurements of temperature and salinity were performed using a SeaBird
SBE 911plus CTD/Carousel system fitted with an in situ fluorometer and 24 Niskin bottles. More
details about the BIOSOPE cruise strategy are given in the Biogeoscience special issue
introductory article by Claustre et al., (2008) while the OUTPACE cruise strategy is detailed in
Moutin et al. (2017). Euphotic layer depths (Ze) were calculated as described in Raimbault et al.
(2008) and Moutin et al. (2018).

**3.3 Inorganic nutrients**

Nutrients were collected in 20 mL PE vials and analyzed directly on a SEAL Analytical auto-
analyzer following Aminot and Kérouel (2007) on board during BIOSOPE and at the laboratory
during OUTPACE from frozen (-20°C) samples.

**3.4 Particulate Organic Carbon (POC)**

Seawater samples (~2 L) were filtered through pre-combusted 25 mm GF/F filters, dried at 60 °C
and stored in 1.5 mL eppendorfs PE tubes. Particulate Organic Carbon (POC) was analyzed on a
CHN elemental analyzer (Perkin Elmer, 2400 series).

**3.5 Total Chlorophyll *a* (TChl*a*)**

For pigment analyses, 2 L of seawater were filtered through 25 mm GF/F filters and stored in liquid
nitrogen and -80°C until processing. Extraction was done in 3 mL 100% methanol, followed by
sonication and clarification by filtration on a new GF/F filter. Extracted pigments (Chl*a* and
fucoxanthin) were then analyzed by HPLC (High Performance Liquid Chromatography) according
to the procedure detailed in Ras et al. (2008).

**3.6 Particulate Biogenic and Lithogenic Silica (BSi/LSi)**

Samples were collected for silicon stocks as particulate biogenic and lithogenic silica (BSi and LSi)
and dissolved orthosilicic acid (Si(OH)$_4$) similarly on both cruises. For BSi/LSi, between 1.5 and
2.5 L Niskin samples were filtered through cascading polycarbonate 47 mm filters. During



BIOSOPE, whole samples were filtered through three cascading filters of 0.2, 2, and 10 µm. During
OUTPACE, the size-fractionation used was 0.4 and 3 µm respectively. Filters were rinsed with 0.2
µm filtered seawater, folded in 4 and placed in Petri dishes and dried overnight at 60°C. Filters
were then stored at room temperature and analyzed in the laboratory. BSi and LSi were measured
using Paasche (1973) as modified by Nelson et al. (1989): BSi and LSi were extracted on the same
filter after successive basic and acid treatments. BSi was extracted during a hot sodium hydroxide
(NaOH 0.2 N) attack (60 min), which converted BSi into the dissolved orthosilicic acid form.
$Si(OH)_4$ was then quantified using the Strickland and Parsons (1972) spectrophotometric method.
After the first basic attack, filters were rinsed free of remaining $Si(OH)_4$ and dried again at 60°C.
LSi, preserved in the sample, was then treated with hydrofluoric acid (HF 2.9 N) for 48 h. In the
same way, LSi was measured through quantification of the dissolved $Si(OH)_4$ form. Precisions for
BSi and LSi measurements were 4 and 6 nmol $L^{-1}$ respectively (twice the standard deviation of
blanks). It has been demonstrated that for coastal samples, significant leaching of orthosilicic acid
from LSi could occur during the first NaOH attack (up to 15 %) (Ragueneau and Tréguer, 1994).
This is particularly the case when high LSi concentrations are present. Kinetic assays of orthosilicic
acid were conducted in some samples from the Marquesas, Gyre, East-Gyre and near Upwelling
stations during BIOSOPE, but results revealed negligible LSi interferences after an extraction time
of 60 min.
Biogenic silica export fluxes were determined from drifting sediment traps deployed at three depths
(153, 328, 519 m) at the three long duration stations of the OUTPACE cruise. Each trap was
deployed for 4 consecutive days, and the average daily flux was quantified by adding the amount
of dissolved Si in each trap to the measured BSi concentration to account for BSi dissolution in the
trap samples during storage. This step proved necessary, as BSi dissolution ranged between 16 and
90 % depending on the samples.
**3.7 Si bulk and specific uptake rates ($\rho$Si/VSi)**
During BIOSOPE, dawn-to-dawn in situ Si uptake experiments were performed using an immersed
production line, at six incubation depths (50 %, 25 %, 15 %, 8 %, 4 % and 1 % light level). Seawater
(275 mL) samples were spiked with 632 Bq of radiolabeled $^{32}$Si-silicic acid solution (specific
activity of 23.46 kBq µg-Si $^{-1}$). For all samples, $Si(OH)_4$ addition did not exceed 0.4 % of the initial
concentration. After incubation, samples were filtered through cascading polycarbonate



membranes (0.2, 2 and 10 µm, 47 mm). Filters were rinsed with filtered (0.2 µm) seawater, and
placed in scintillation vials. The $^{32}$Si uptake was measured in a Packard 1600-TR scintillation
counter by Cerenkov effect, following the method described by Tréguer and Lindner (1991) and
Leynaert (1993). Precision of the method averages 10 % to 25 % for the less productive station.

**3.8 Si uptake kinetics**

Samples used were collected from the same Niskin bottles as those used for in situ incubation at
the depth of the Chl*a* maximum. Six samples from each depth received non-radioactive Si(OH)$_4$
additions so that concentrations increased respectively by 0, 1.1, 2.3, 4.5, 13.6, 36.4 µM. Bottles
were incubated on board in a deck incubator for 8h using neutral nickel screens. Samples were
thereafter treated as described for in situ samples. Kinetic parameters Ks and Vmax were calculated
by fitting the data to a hyperbolic curve using the Sigmaplot® hyperbola fit.

**3.9 Siliceous phytoplankton determinations**

Seawater samples were preserved with acidified Lugol's solution and stored at 4ºC. A 500 mL
aliquot of the sample was concentrated by sedimentation in glass cylinders for six days. Diatoms
were counted following the method described by Gomez et al. (2007).

**3.10 Phytoplankton net samples**

During the OUTPACE cruise, additional phyto-net hauls were undertaken at each site integrating
the 0-150 m water column, except at stations LD-C, 14 and 15 where they integrated the 0-200 m
water column due to the presence of a very deep Deep Chlorophyll *a* Maximum (DCM). Samples
were preserved in acidified lugol, and observed in a Sedgewick-rafter chamber. A semi-quantitative
species list (dominant, common, rare) was established.

**4 Results**

**4.1 Hydrological systems and nutrient availability**

The hydrological structures crossed during the two transects have been carefully detailed in
companion papers (Claustre et al., 2008; Moutin et al., 2018; Fumenia et al., 2018) and will not be
presented in detail here. For the sake of clarity in the present article, main hydrological systems are
described as follows. During the BIOSOPE cruise, five main hydrological systems were defined



from West to East: the HNLC system comprising long duration (LD) stations MAR (Marquesas)
and HNL and station 1; the South Tropical Pacific (STP) system from stations 2 to 6; the central
part of the South Pacific Gyre (SPG) from station 7 to 13 including the LD station GYR; the Eastern
Gyre HNLC area from stations 14 to 19 including LD station EGY (Eastern Gyre); and the coastal
Peru-Chile Upwelling system from station 20 to 21 including LD stations UPW and UPX. During
OUTPACE, two main systems were encountered, from West to East, the MA (Melanesian
Archipelago) from stations 1 to 12 and including LD stations A and B, and the South Pacific Gyre
(SPG) from stations 13 to 15 and including LD station C.
During both cruises, eutrophic to ultra-oligotrophic conditions were encountered. During
OUTPACE, $Si(OH)_4$ concentrations were <1 μM at all stations in the surface layer, with values as
low as 0.3-0.6 μM at 5 m depth at certain stations (Fig. 2). The 1 μM isoline was centered at ~100
m in the western part of the MA, and deepened to ~200 m in the SPG. Concentrations at 300 m
were quite low (<2 μM) over the entire transect. Nitrate concentrations were similarly depleted in
the surface layer, with values <0.05-0.1 μM in the first 80 m in the western part of the MA (until
station 6), which deepened to 100 m over the rest of the transect. Yet nitrate concentrations
increased with depth more rapidly than orthosilicic acid, reaching concentrations close to 7 μM at
300 m depth.
Phosphate was below detection limits in the western part of the MA (stations 1 to 11, and station
B) over the first 50 m, but increased to values comprised between 0.1 and 0.2 μM in the SPG.
Concentrations only increased to 0.6-0.7 μM at 300 m depth.
During BIOSOPE, both the nitracline and phosphacline extended very deeply (~200 m) in the
regions of the STP, SPG and Eastern Gyre (Fig. 3). They surfaced at both ends of the transect in
the upwelling system and near the Marquesas Islands, but contrary to nitrate which was severely
depleted, phosphate was never found <0.1 μM in the surface layer (except at the subsurface at site
14). The distribution of orthosilicic acid concentrations were less clearly contrasted, with general
surface values comprised between 0.5 and 1 μM in the surface layer, except in the western part of
the transect from station 1 to the GYR station, and in the upwelling system, where concentrations
were > 1 μM and up to 8.9 μM at the surface and increasing rapidly with depth.



**4.2 Total Chl*a* and fucoxanthin distribution**
Total Chl*a* (TChl*a*) distributions are presented for both cruises along longitudinal transects together
with fucoxanthin concentrations, a diagnostic pigment for diatoms (Fig. 4a, b). During OUTPACE,
the Melanesian Archipelago system was clearly enriched in TChl*a* compared to the South Pacific
Gyre and showed non-negligible concentrations in surface layers as well as a pronounced DCM
reaching up to 0.45 µg L$^{-1}$ at station 11. The observed DCM progressively deepened eastwards,
from 70 m depth at LD-A to 108 m at station 12. The DCM depth generally closely followed the
euphotic layer depth ($Z_{eu}$) or was located just below it. The highest surface concentrations were
found at stations 1 to 6, between New Caledonia and Vanuatu (0.17 to 0.34 µg L$^{-1}$) while the SPG
surface water stations showed a depletion in Chl*a* (0.02 to 0.04 µg L$^{-1}$). A DCM subsisted in this
region, but was observed to be deeper (125 to 150 m) and of lower amplitude (0.17 to 0.23 µg L$^{-1}$)
than in the MA region. Fucoxanthin concentrations closely followed the DCM, but were extremely
low over the entire transect, with a maximum concentration of 17 ng L$^{-1}$ in the MA and of 4 ng L$^{-1}$
$^{1}$ in the SPG.
The BIOSOPE cruise evidenced a very similar Chl*a* distribution in the central SPG than during the
OUTPACE cruise, with extremely low surface concentrations and a very deep Chl*a* maximum
located between 180 - 200 m ranging between 0.15 and 0.18 µg L$^{-1}$. On both sides of the central
SPG, the DCM shoaled towards the surface at the MAR station at the western end of the transect
(0.48 µg L$^{-1}$ at 30 m) and at the UPW station at the eastern end of the transect (3.06 µg L$^{-1}$ at 40
m). Fucoxanthin concentrations did not exceed 9 ng L$^{-1}$ at any station between the STP and the
Eastern Gyre (between LD-HNL and station 17), thus showing ranges similar to the OUTPACE
cruise measurements. Fucoxanthin increased moderately at the MAR station (85 ng L$^{-1}$), while it
peaked in the Peru-Chile upwelling system with concentrations reaching 1,595 ng L$^{-1}$ at LD-UPW
but remained much lower at the LD-UPX station (200 ng L$^{-1}$).
**4.3 Total and size-fractionated Biogenic and Lithogenic Silica standing stocks**
Total Biogenic silica (BSi) concentrations were extremely low during the OUTPACE cruise (Fig.
5a) and ranged between 2 and 121 nmol L$^{-1}$ in the surface layers, with an average concentration of
17 nmol L$^{-1}$. Similarly to TChl*a* and fucoxanthin, the highest BSi levels were encountered over the
MA, with peak values mostly found at the surface, at stations 1 and 2 and from stations 4 to 7, and
with very moderate increases at depth (stations 5 and 10). The average BSi concentration decreased



from 20 to 8 nmol L$^{-1}$ from the MA to the SPG. In the SPG, maximum BSi levels were found at
the DCM, between 125 and 150 m. Total Lithogenic Silica (LSi) concentrations were measured in
a very similar range (Fig. 5b), between 2 and 195 nmol L$^{-1}$, with a peak value at station 2 at 100 m.
Also, LSi was ranged from 5 to 30 nmol L$^{-1}$ over the transect, with highest values observed close
to 100 m, while averaged concentrations followed the same trend as BSi, decreasing from 16 to 9
nmol L$^{-1}$ between the MA and the SPG.
During the BIOSOPE cruise, three main regions could be differentiated: a first region covering the
ultra-oligotrophic central area from station 1 to station 20, where average BSi concentrations were
as low as 8 nmol L$^{-1}$ (Fig. 5c). At the western end of the transect, the first three stations in the
vicinity of the Marquesas Islands had higher concentrations with average values of 104 nmol L$^{-1}$.
The eastern end of the transect, located in the Peru-Chile Upwelling system, displayed much higher
and variable values, averaging 644 nmol L$^{-1}$, with a maximum concentration of 2,440 nmol L$^{-1}$ at
the UPW station at 60 m. At both ends of the transect, siliceous biomass was mainly distributed in
the upper 100 m. Lithogenic silica followed the same trends (Fig. 5d), with extremely low values
over the central area (average of 7 nmol L$^{-1}$) with a few peaks close to 30 nmol L$^{-1}$ (stations 12 and
EGY). LSi was again higher at both ends of the transect but with less amplitude than BSi, with
average values of 26 nmol L$^{-1}$ close to the Marquesas, and of 57 nmol L$^{-1}$ in the coastal upwelling
system. The maximum values close to 150 nmol L$^{-1}$ were associated to the BSi maximums at the
UPW sites.
Size-fractionated integrated BSi stocks were calculated for both cruises over the 0-125 m layer,
except for the BIOSOPE cruise at station UPW1, which was only integrated over 50 m and at
stations UPX1 and UPX2 which were integrated over 100 m (Fig. 6a, b, Appendix 1). Total BSi
stocks were similarly very low in the ultra oligotrophic central gyre and averaged 1 mmol Si m$^{-2}$
during both cruises. During BIOSOPE, the stocks measured closed to the Marquesas averaged 9.85
mmol Si m$^{-2}$ (with a peak of 24.12 mmol Si m$^{-2}$ at the MAR station). On the eastern end of the
transect, stocks increased to a peak value of 142.81 mmol Si m$^{-2}$ at the UPW2 station and averaged
65.68 mmol Si m$^{-2}$ over the coastal upwelling system. Size-fractionation was only carried out at
the long duration stations, but showed an overall non negligible contribution of the pico-sized
fraction (0.2-2 µm) to BSi standing stocks of 11 ± 9 %. This contribution of the pico-size fraction
to integrated siliceous biomass was highest at the GYR, EGY and UPX1 stations reaching 25, 18
and 24 % respectively.





During OUTPACE, integrated BSi stocks ranged between 1.25 and 4.11 mmol Si m$^{-2}$ over the MA,
and decreased to 0.84 to 1.28 mmol Si m$^{-2}$ over the SPG (Fig. 6c, Appendix 2). Here, size-
fractionation was conducted at all sites and the contribution of the 0.4 - 3 µm, which will be
assimilated to the pico-size fraction hereafter, was higher than during BIOSOPE, with an average
contribution of 26 ± 12 %. The importance of the picoplanktonic Si biomass was higher in the SPG
(36 ± 12 %) than over the MA (22 ± 10 %).

**269    4.4 Si uptake rates and kinetic constants**

Si uptake rate measurements using the $^{32}$Si radioactive isotope were only conducted during the
BIOSOPE cruise. The same size-fractionation was applied to production and kinetic experiment
samples. Vertical profiles of gross production rates (ρSi) confirm the previous stock information
and show that the most productive stations, in decreasing order of importance, are the UPW, UPX
and MAR stations (Fig. 7a), with 1.98, 1.19 and 0.22 µmol Si L$^{-1}$ d$^{-1}$ at 10 m respectively. Si uptake
rates remained below 0.015 µmol Si L$^{-1}$ d$^{-1}$ at central HNLC and oligotrophic stations HNL, EGY
and GYR. Si uptake rates in the picoplanktonic size fraction showed similar trends (Fig. 7b),
despite higher values at UPX (0.076 µmol Si L$^{-1}$ d$^{-1}$) than at UPW (0.034 µmol Si L$^{-1}$ d$^{-1}$). Uptake
rates in that size fraction were intermediate at the MAR station with maximum value of 0.005 µmol
Si L$^{-1}$ d$^{-1}$, while it remained below 0.001 µmol Si L$^{-1}$ d$^{-1}$ at the central stations. Specific Si uptake
(VSi normalized to BSi) rates for the picoplanktonic size fraction were even more elevated and
reached maximum values of 3.64, 1.32, 0.75, 0.37 and 0.14 d$^{-1}$ at the UPW, UPX, HNL, EGY and
MAR stations respectively. Total specific Si uptake rates were extremely high in the coastal
upwelling system, with values of 2.57 and 1.75 d$^{-1}$ at UPX and UPW respectively, and lower but
still elevated values at the MAR station (0.75 d$^{-1}$). VSi at the central stations (HNL, EGY, GYR)
were moderate to low and ranged between 0.02 and 0.24 d$^{-1}$.
Total ΣρSi reached 52.4 mmol Si m$^{-2}$ d$^{-1}$ at UPW2 station, an order of magnitude higher that the
rate measured at the MAR station (5.9 mmol Si m$^{-2}$ d$^{-1}$) and 3 orders of magnitude higher than at
EGY, where the lowest value was obtained (0.04 mmol Si m$^{-2}$ d$^{-1}$). Integrated picoplanktonic Si
uptake rates (ΣρSi for 0.2-2 µm) were highest at both upwelling stations (Table 1), followed by the
MAR station. The relative average contribution of the picoplanktonic size fraction to total Si uptake
rates was highest at the central stations (32 % at GYR, 19 % at EGY and 11 % at HNL) while it
was lowest on both ends of the transect (5 % at MAR, and 3 and 7 % at UPW and UPX stations).





Si uptake kinetic experiments were conducted at some long duration stations at the surface and/or
depth of the DCM depending on the location of biomass. Results for the picoplanktonic fraction
clearly indicate an active biological uptake (Fig. 8), generally following hyperbolic uptake kinetics.
The hyperbolic curve fitting failed for only 2 out of the 8 kinetic uptake experiments performed on
the 0.2-2 µm size-fraction (at the DCM at the HNL station and at the surface at the UPX station).
Maximum theoretical specific uptake rates ($V_{max}$) values were high, ranging from 1.9 $d^{-1}$ at the
MAR station to 6.1 $d^{-1}$ at the surface at the UPX station. Half-saturation constants ($K_S$) were also
elevated ranging from 5.4 µM at the MAR station to as much as 38.3 µM at the UPX station and
in all cases much higher than ambient $Si(OH)_4$ concentrations.
**4.5 Diatom distribution and community structure**
Microscopical examinations confirmed the presence of diatoms at every station during both cruises.
Diatoms were found in very low abundances during the OUTPACE cruise and only reached
maximum values of 20,000-30,000 cells $L^{-1}$ on two occasions, at stations LD-B at the surface and
at station 5 at the DCM (Fig. 9a). Mean diatom concentrations in the MA at the surface were 4,440
$\pm$ 7,650 cells $L^{-1}$ while at the DCM, mean concentrations were about 2-fold lower (2,250 $\pm$ 4,990
cells $L^{-1}$). Diatom abundance decreased dramatically in the SPG, with values as low as 25 $\pm$ 19
cells $L^{-1}$ at the surface layers and 145 $\pm$ 54 cells $L^{-1}$ at the DCM. The richness of diatoms was higher
in the MA than in the SPG, with an average number of taxa of respectively 9 $\pm$ 4 and 2 $\pm$ 1 in the
surface layer (Fig. 9b). The richness increased at the DCM level, with 12 $\pm$ 8 taxa in the MA and
5 $\pm$ 1 taxa in the SPG. Diatom contribution to biomass was accordingly extremely low and remained
below 3 % (Fig. 9c). The diatom contribution to C biomass increased more significantly only at
two stations: at station LD-B (9 % at the surface) and at station 5 where the maximum value for
the cruise was observed (11.5 % at the DCM).
During BIOSOPE, the central stations showed a record low diatom abundance with less than 100
cells $L^{-1}$ from stations 2 to EGY (Fig. 10). The eastern part of the SPG and the HNL stations were
characterized by slightly higher abundances (from 100 to 1,000 cells $L^{-1}$), followed by the UPX
station, where abundances were similar to the MAR station at the surface (~25,000 cells $L^{-1}$).
Highest abundances were observed at the UPW, with bloom values of 256,000 cells $L^{-1}$ on average
(with a peak abundance of 565,000 cells $L^{-1}$ at the surface). Similar results compared to OUTPACE
showed an extremely low richness at all central stations (data not shown) with on average 3 $\pm$ 2



diatom taxa, while richness increased at the western HNLC region with 13 ± 4 taxa at the MAR
and HNL stations. Richness was highest at the UPW station with 20 ± 4 taxa and decreased again
at the UPX station (5 ± 3).
The dominant diatom species for each system sampled over the course of the two cruises are
summarized in Table 1 and Appendix 3. During OUTPACE, very similar species were encountered
in both regions and were mainly dominated by pennate species such as *Pseudo-nitzschia* spp., *P.*
*delicatissima*, *Cylindrotheca closterium* and *Mastogloia woodiana*. However, Diatom-Diazotroph
Associations (DDAs) such as *Rhizosolenia styliformis*, *Climacodium frauenfeldianum* and
*Hemiaulus hauckii* were more abundantly found in the MA. Other siliceous organisms such as
radiolaria were also more abundant in the SPG and at LD-B than in the MA (Appendix 3). Overall
microplanktonic diazotroph abundance were much higher over the MA than in the gyre, with a
predominance in plankton nets of *Trichodesmium*, *Richelia intracellularis* (alone or in DDAs),
*Crocosphaera* and other filamentous cyanobacteria such as *Katagnymene* (Appendix 3).
Diatom community structure for the BIOSOPE cruise has already been discussed extensively in
Gomez et al. (2007). In summary, the stations characterized by medium diatom abundances such
as MAR, HNL, 18, 20 and EGY (Fig. 10) were mainly dominated by the pennate diatom *Pseudo-*
*nitzschia delicatissima* in particular at the MAR station, where it represented on average 90 % of
all diatoms over the 0-100 m layer. Extremely low abundance stations (< 200 cells L$^{-1}$) from the
middle of the SPG (stations 2 to 14) did not show any consistent community, with varying dominant
species across stations and along vertical profiles as well. Maximum abundances at these sites were
consistently found at depth, between 100 and 200 m. In the Peru-Chile upwelling, diatom
community structure was mostly dominated by small and colonial centric species such as
*Chaetoceros compressus* and *Bacteriastrum* spp. at the UPW station where abundances were
highest (565,000 cells L$^{-1}$) and such as *Skeletonema* sp. and *Thalassiosira anguste-lineata* at the
UPX station where abundances decreased to 10,000-40,000 cells L$^{-1}$. In this system, the highest
abundances were found in the first 10 m.
**4.6 Si export fluxes**
Particulate silica export fluxes were measured from drifting trap deployments at each long duration
station during OUTPACE and are presented in Table 3. BSi daily export fluxes below the mixed
layer at 153, 328 and 529 m were extremely low at all sites, with lowest values at site A (0.5 to 0.1



µmol Si m$^{-2}$ d$^{-1}$), highest at site B (3 to 5 µmol Si m$^{-2}$ d$^{-1}$) and intermediate at site C (0.5 to 2 µmol
Si m$^{-2}$ d$^{-1}$).

## 5 Discussion

### 5.1 Si budgets for the South Pacific

In the following section, values from previous studies are compared (Table 4) with the results
obtained across this under-studied region of the Pacific Ocean, which is characterized by the most
oligotrophic and Chl*a* depleted waters worldwide (Ras et al., 2008). On one hand, size-fractionated
biomass and export fluxes were obtained during the OUTPACE program, while on the other hand,
size-fractionated production and biomass budgets were quantified during the BIOSOPE program.
Regarding values obtained at both ends of the BIOSOPE transects, i.e. in the Peru-Chile upwelling
system and in the HNLC system surrounding the Marquesas Islandss, Σ$\rho$Si rates compare well with
previous studies from other similar regions (Table 4). Integrated Si production rates at the UPW
stations are in the middle range (42-52 mmol Si m$^{-2}$ d$^{-1}$) of what was previously found in coastal
upwellings. Values are however almost double to what was previously observed in the Peru
upwelling by Nelson et al. (1981), although less productive than the Monterey Bay and Baja
Californian upwelling systems (Nelson and Goering, 1978; Brzezinski et al., 1997). For oceanic
HNLC areas, values obtained (0.8 to 5.6 mmol Si m$^{-2}$ d$^{-1}$) cover the range of rates measured in
HNLC to mesotrophic systems of the North Atlantic, Central Equatorial Pacific and Mediterranean
Sea. However, integrated rates obtained for the oligotrophic area of the South Eastern Pacific Gyre
are to our knowledge among the lowest ever measured. Indeed, values range from 0.04 to 0.20
mmol Si m$^{-2}$ d$^{-1}$, they are thus lower than average values previously measured at BATS and
ALOHA stations (0.42 and 0.19 mmol Si m$^{-2}$ d$^{-1}$ respectively) (Brzezinski and Kosman, 1996;
Nelson and Brzezinski, 1997; Brzezinski et al., 2011). However, they are similar to measurements
performed in autumn (0.04-0.08 mmol Si m$^{-2}$ d$^{-1}$) in a severely Si-limited regime of the North
Atlantic (Leblanc et al., 2005b). Previous studies have evidenced limitation of diatom Si production
by Si (Leynaert et al., 2001), but more recently evidence of co-limitation by both Si and Fe was
found in the central Equatorial Pacific (Brzezinski et al., 2008). This would be a more than likely
scenario for the SPG, given the very low silicic acid (Fig.2 & 3) and Fe concentrations (0.1 nM
and ferricline below 350 m depth, Blain et al., 2008) measured during both cruises. The
approximate surface area of mid-ocean gyres was estimated to be 1.3 x 10$^8$ km$^2$ (representing




approximately 1/3 of the global ocean) yielding a global contribution of only 26 Tmol Si y$^{-1}$ gross
silica production, i.e. approximately 9-13% of the budget calculated for the global ocean of 240
Tmol Si y$^{-1}$ according to Nelson et al. (1995). This budget has been recently revised down to 13
Tmol Si y$^{-1}$ reducing the contribution of subtropical gyres to 5-7% of global marine silica
production (Brzezinksi et al., 2011; Tréguer and de La Rocha, 2013). However, the range provided
in Nelson et al. (1995) in the calculation of their global Si production fluxes for mid-ocean gyres
was of 0.2 – 1.6 mmol m$^{-2}$ d$^{-1}$. Our values would, once again, lower the contribution of these vast
oceanic regions to global Si production, although the present data is only based on two production
station measurements and warrants further measurements for this region. Nevertheless, it can be
expected that the most ultra-oligotrophic region of the world ocean would contribute even less to
total Si production than the other oligotrophic systems listed in Table 4 and that in particular, the
Si production in the ultra-oligotrophic Southern Tropical Gyre would be lower than the Northern
Tropical Gyre.
Integrated Si biomass also reflects the very low contribution of diatoms in this system, which was
more than 2-fold lower in in the South Pacific Gyre than in the Melanesian Archipelago (Table 5).
In the SPG, the lowest Si stocks were measured (~1 mmol Si m$^{-2}$), and were similar to lower-end
values found in the ultra-oligotrophic Eastern Mediterranean Basin in autumn and in other
oligotrophic areas of the North Pacific Subtropical Gyre and of the Sargasso Sea (Table 5 and
references therein). It is probable that ΣρSi production and BSi stocks could have been slightly
higher less than a month earlier in the season on the western part of the OUTPACE transect in the
MA. Indeed, the satellite-based temporal evolution of Chl*a* at stations LD-A and LD-B showed
decreasing concentrations at the time of sampling (de Verneil et al., 2018), while the situation did
not show any temporal evolution for the SPG, thus suggesting that the biogenic silica budget for
this area is quite conservative under a close to steady-state situation.
Lastly, our Si export flux measurements by drifting sediment traps are the lowest ever measured
and are about two orders of magnitude lower than those from other oligotrophic sites such as BATS
in the Atlantic or ALOHA in the Pacific Ocean (Table 6). They represent a strongly negligible
fraction of surface Si stocks, implying no sedimentation at the time of sampling, and that active
recycling and grazing occurred in the surface layer. Indeed, surface temperatures higher than 29°C
at all long duration sites, may favor intense dissolution in the upper layer, while active zooplankton
grazing was also documented, removing between 3 and 21% of phytoplankton stocks daily (Carlotti



et al., 2018). The virtual absence of silica export from the surface layer well agrees with the
conclusion of Nelson *et al.* (1995) that no siliceous sediment is accumulating beneath the central
ocean gyres.

**5.2 Siliceous plankton community structure in the South Tropical Pacific**
The main feature observed during OUTPACE was a bi-modal distribution of diatom communities,
either at the surface and/or at the DCM level depending on stations, which deepened towards the
East, following the increasing oligotrophy gradient, similarly to what was previously described in
the Mediterranean Sea (Crombet et al., 2011). A similar feature, showing a particularly deep DCM,
up to 190 m in the SPG at 1.2-fold the euphotic depth (Ras et al., 2008), was observed during
BIOSOPE, revealing a known strategy for autotrophic plankton cells in nutrient depleted waters to
stay at the depth where the best light vs nutrient ratio is obtained (Quéguiner, 2013).
If the presence of DCMs in oligotrophic mid-ocean gyres are well known, associated to the
dominance of small pico-sized phytoplankton (Chavez et al., 1996), studies documenting
phytoplankton community structure in the South Tropical Pacific Ocean, an area formerly called a
« biological desert », are still very scarce. In the review of planktonic diatom distribution by
Guillard and Kilham (1977) referencing biocenoses for all main oceanic water bodies and for which
thousands of articles were processed, the diatom composition for the South Tropical region was
referred to as « No species given (flora too poor) ». Since then only a few studies mentioning
phytoplankton community structure, mostly located along the equator were published, such as
Chavez et al. (1990); Chavez et al. (1991); Iriarte and Fryxell (1995); Kaczmarska and Fryxell
(1995); and Blain et al. (1997). In Semina and Levashova (1993) some biogeographical distribution
of phytoplankton including diatoms is given for the entire Pacific region, yet the Southern tropical
region is limited to more historical Russian data and rely on very few stations. The only diatom
distribution for the South Tropical Gyre was published for the present data set by Gomez et al.
(2007) in the BIOSOPE special issue. Hence the present data contributes to documenting a severely
understudied, yet vast area of the world ocean.
The oceanic regions covered during both cruises may be clustered into three main ecological
systems with relatively similar diatom community structures: the nutrient-rich coastal upwelling
system near the Peru-Chile coast, where diatom concentrations exceeded 100,000 cells L$^{-1}$, the Fe-
fertilized areas of the Melanesian Archipelago and West of Marquesas Islands, where





445 concentrations could locally exceed 10,000 cells $L^{-1}$, and all the other ultra-oligotrophic regions

446 (mainly the South Pacific Gyre system) characterized by extremely low diatom abundances,

447 usually <200 cells $L^{-1}$.

448 The upwelling area was characterized by a distinct community, not found in the other regions,

449 composed of typical neritic and centric colonial species such as *Skeletonema* sp., *Bacteriastrum*

450 spp., *Chaetoceros compressus*, *Thalassiosira subtilis* and *T. anguste-lineata*. These first three

451 species were already documented as abundant in the Chile upwelling by Avaria and Munoz (1987),

452 whereas *T. anguste-lineata* was reported along the Chilean coast from 20°S to 36°S (Rivera et al.,

453 1996) and was also documented in the upwelling system West of the Galapagos Islands (Jimenez,

454 1981). The highest ρSi production values were measured at the offshore UPW station where

455 *Bacteriastrum* spp. and *Chaetoceros compressus* co-occurred as the two dominant species, whereas

456 ρSi rates were halved at the closest coastal station UPX, associated to lower abundances of diatoms,

457 with co-occurring dominance by *Skeletonema* sp. and *Thalassiosira anguste-lineata*.

458 The HNLC regions off the Marquesas Islands (MAR) and in the Eastern Gyre (stations 14-20,

459 BIOSOPE) and the oligotrophic region (N-deprived but Fe-fertilized region of the MA), with

460 bloom situations at stations 5 and LD-B (OUTPACE), showed strong similarities in terms of

461 diatom community structure and were all mainly dominated by the medium-sized pennate diatoms

462 of the *Pseudo-nitzschia delicatissima/subpacifica* species complex. These pennate species are

463 commonly reported for the Central and Equatorial Pacific Ocean (Guillard and Kilham, 1977;

464 Iriarte and Fryxell, 1995; Blain et al., 1997). During BIOSOPE, *Pseudo-nitzschia delicatissima*

465 were often seen forming « needle balls » of ~100 µm diameter which suggests an anti-grazing

466 strategy from micro-grazers (Gomez et al., 2007), a strategy already described by several authors

467 (Hasle, 1960; Buck and Chavez, 1994; Iriarte and Fryxell, 1995). Predominance of pennate diatoms

468 over centrics has previously been observed in the N-depleted environment of the Equatorial Pacific

469 (Blain et al., 1997; Kobayashi and Takahashi, 2002), and could correspond to an ecological

470 response to diffusion-limited uptake rates, favoring elongated shapes, as suggested by Chisholm

471 (1992). Furthermore, net samples from the OUTPACE cruise showed a numerically dominant

472 contribution of *Cylindrotheca closterium* over 0-150 m at most stations of the MA (Appendix 3),

473 with a strong dominance at LD-B, even though their contribution to biomass is minor given their

474 small size. *Pseudo-nitzschia* sp. and *Cylindrotheca closterium* have been shown to bloom upon Fe-

475 addition experiments (Chavez et al., 1991; Fryxell and Kaczmarska, 1994; Leblanc et al., 2005a;



Assmy et al., 2007) and may reflect the significantly higher dissolved Fe concentrations measured
in the MA (average 1.9 nM in the first 100 m) compared to the SPG (0.3 nM) (Guieu et al., in rev).
In the Equatorial Pacific, Fe-amendment experiments evidenced the rapid growth of *Cylindrotheca*
*closterium*, with a high doubling rate close to 3 $d^{-1}$ (Fryxell and Kaczmarska, 1994), which can
explain why this species is often numerically dominant.
Fast growing colonial centric diatoms such as *Chaetoceros* spp. were notably absent from the MA,
except at stations 5 and LD-B, where mesoscale circulation increased fertilization (de Verneil et
al., 2018) and allowed a moderate growth (observed in both Niskin samples and net hauls),
resulting in an increased contribution of diatoms to total C biomass of approximately 10% (Fig.
9c). Other typical bloom species such as *Thalassiosira* spp. were completely absent from the
species from the Niskin samples but observed at low abundance in some net haul samples.
Nonetheless, very large centrics typical of oligotrophic waters such as *Rhizsolenia calcar-avis*
(Guillard and Kilham, 1977) were present in low numbers at all stations and in all net hauls, and
represented a non-negligible contribution to biomass despite their low abundance.
One difference with the N-replete Marquesas HNLC system was that the hydrological conditions
of the MA were highly favorable for the growth of diazotrophs, with warm waters (>29°C),
depleted N in the surface layer associated to high Fe levels, while P was likely the ultimate
controlling factor of N-input by $N_2$-fixation in this region (Moutin et al., 2008; Moutin et al., 2018).
$N_2$-fixation rates were among the highest ever measured in the open ocean during OUTPACE in
this region (Bonnet et al., 2017), and the development of a mixed community, composed of
filamenteous cyanobacteria such as *Trichodesmium* spp. and other spiraled-shaped species,
unicellular diazotrophs such as UCYN, *Crocosphaera watsonii*, and Diatom-Diazotroph
Associations (DDAs) was observed (Appendix 3). The highest rates were measured at the surface
at stations 1, 5, 6 and LD-B (Caffin et al., this issue) and the major contributor to $N_2$-fixation in
MA waters was by far *Trichodesmium* (Bonnet et al., 2018). In the Niskin cell counts, DDAs known
to live in association with the diazotroph *Richelia intracellularis* such as *Hemiaulus hauckii,*
*Chaetoceros compressus* and several species of *Rhizosolenia* such as *R. styliformis, R. bergonii, R.*
*imbricata* and the centric *Climacodium frauenfeldianum* known to harbor a genus related to
*Cyanothece* sp. (Carpenter, 2002) were all found in low abundance in the water sample cell counts,
contributing to less than 1% of total diatoms. Exceptions were observed at sites 1 and 2 where their
contributions increased to 2.3 and 8% respectively. The low contribution of DDAs to the



diazotrophs community was confirmed by direct cell counts and nifH gene sequencing (Stenegren
et al., 2018). Notably, the presence of *Richelia intracellularis* was not observed in the Niskin lugol-
fixed water samples, but *Rhizosolenia styliformis* with *Richelia*, and some isolated *Richelia* cells
were observed abundantly in net hauls. The latter were found to be dominant at stations 1 and LD-
B, where the highest fixation rates were measured. *Richelia*, alone or in association with *R.*
*styliformis* were much less abundant in the South Pacific Gyre, where Fe is prone to be the limiting
nutrient for $N_2$-fixation rates despite higher P availability, pointing to less favorable growth
conditions for diazotrophs. Yet, the overall dominance of *Trichodesmium, Crocosphaera* and other
filamenteous cyanobacteria (Appendix 3) in the net samples reveals that DDAs were very minor
contributors to $N_2$-fixation during OUTPACE. This was also evidenced through NanoSIMS
analyses (Caffin et al., 2018). In order to explain the growth of diatoms in this severely N-depleted
region, one can quote the use of diazotroph-derived nitrogen (DDN), i.e. the secondary release of
$N_2$ fixed by diazotrophs, which showed to be efficiently channeled through the entire plankton
community during the VAHINE mesocosm experiment (Bonnet et al., 2016). In this latter study
off shore New Caledonia, *Cylindrotheca closterium* grew extensively after a stimulation of
diazotrophy after P-addition in large volume in situ mesocosms in New Caledonia (Leblanc et al.,
2016). As previous studies had already observed a co-occurrence of elevated *C. closterium* with
several diazotrophs (Devassy et al., 1978; Bonnet et al., 2016), this recurrent association tends to
confirm our previous hypothesis of a likely efficient use of DDN released as $NH_4$ by this fast
growing species (Leblanc et al., 2016). This could be another factor, besides Fe-availability,
explaining its success. A similar hypothesis may be invoked for the presence of *Mastogloia*
*woodiana*, a pennate diatom known to be occasionally dominant in the North Pacific Subtropical
Gyre blooms (Dore et al., 2008; Villareal et al., 2011). It is also a characteristic species of
oligotrophic areas (Guillard and Kilham, 1977), often observed in association with other DDAs,
which could similarly benefit from secondary N-release (Villareal et al., 2011; Krause et al., 2013).
Lastly, the ultra-oligotrophic region of the SPG investigated both during OUTPACE and BIOSOPE
revealed a base-line contribution of diatoms with often less than 200 cells $L^{-1}$ at the DCM and close
to zero at the surface. In addition, a dominance of small and large pennate species was observed,
such as *Nitzschia bicapitata*, *Pseudo-nitzschia delicatissima*, *Thalassiothrix longissima,*
*Thalassionema elegans* and *Pseudoeunotia* sp., that have already been documented for the
Equatorial Pacific by Guillard and Kilham (1977). Occasional occurrences of some emblematic



species of oligotrophic regions were also observed, such as *Chaetoceros dadayi*, *C.*
*tetrastichon* or *Planktoniella sol*. It can be noted that radiolarians were also more abundant and
more diverse in the ultra-oligtrophic SPG during OUTPACE than in the MA, while unfortunately
no information regarding radiolarians is available for the BIOSOPE cruise.
**5.3 Evidence for active Si uptake in the pico-planktonic size-fraction in the South Tropical**
**Pacific**
The pico-size fraction (<2-3 µm) represented on average 11% of BSi stocks during BIOSOPE, and
26% of BSi stocks during OUTPACE (Fig. 6), which is a non-negligible contribution. If the
importance of pico-size fraction in the BSi stock could be explained by detrital components, its
contribution to $Si(OH)_4$ uptake during BIOSOPE was really surprising but could be explained in
the light of new findings. Indeed, recent studies have evidenced that the pico-phytoplanktonic
cyanobacteria *Synechococcus* can assimilate Si (Baines et al., 2012; Ohnemus et al., 2016; Krause
et al., 2017; Brzezinski et al., 2017), which could explain why Si stocks were detected in this size
fraction. The first hypothesis was to consider broken fragments of siliceous cells passing through
the filter or interferences with lithogenic silica, but these hypotheses were invalidated during
BIOSOPE when Si uptake measurements using $^{32}$Si were also carried out on this pico-size fraction
and revealed a non-negligible uptake, mainly in the Chilean upwelling systems (Fig. 7). It is also
excluded that some broken parts of active nano-planktonic diatoms labelled with $^{32}$Si could have
passed through the filters because of breakage during filtration, as a kinetic type response was
observed in most samples (Fig. 8), implying truly active organisms in the 0.2-2 µm size fraction.
Our results are thus in line with previous findings, as no other organisms below 2-3 µm are known
to assimilate Si, except some small size Parmales, a poorly described siliceous armored planktonic
group which span over the 2-10 µm size class, such as *Tetraparma* sp. (Ichinomiya, 2016), or small
nano-planktonic diatoms such as *Minidiscus* (Leblanc et al., 2018), close to the 2 µm limit (Fig. 11
a,b). The latter two species could occur in the 2-3 µm size-fraction, but are very easily missed in
light microscopy and require SEM imaging or molecular work for correct identification. Presence
of Parmales or nano-planktonic diatoms may explain the measurement of BSi in this 0.4 – 3 µm
size-class for the OUTPACE cruise, but can be excluded as responsible for the Si uptake measured
during BIOSOPE on filters below 2 µm. Rather, during OUTPACE, NanoSIMS imaging revealed
that cytometrically sorted *Synechococcus* cells accumulated Si (Fig. 11c), confirming their
potential role in the Si cycle in the South Tropical Gyre.





According to Baines et al. (2012), the Si content of *Synechococcus*, in some cases, could exceed
that of diatoms, but these authors suggested that they might exert a larger control on the Si cycle
in nutrient-poor waters where these organisms are dominant. In the present study, the largest
contribution of the pico-size fraction to absolute $\Sigma\rho Si$ uptake rates occurred at both ends of the
transect in the Peru-Chile upwelling region and at the MAR station (Table 1), locations which also
corresponded to the highest concentrations of *Synechococcus* observed (Grob et al., 2007).
However, compared to diatoms, this only represented 1 to 5 % of total $\Sigma\rho Si$ uptake, which is
probably not likely to drive the Si drawdown in this environment. This low relative contribution to
$\Sigma\rho Si$ was similarly found at the other end of the transect at HNL and MAR station, but where
absolute uptake rates were moderate. The largest contribution of the pico-size fraction was
measured in the SPG (GYR and EGY sites), where despite very low $\rho Si$ values, the relative $\Sigma\rho Si$
uptake between 0.2 and 2 µm reached 16 to 25 %. Station GYR as well as stations 13 to 15 are
areas that are highly depleted in orthosilicic acid, with concentrations <1 µM from the surface to
as deep as 240 m. Hence, it is probable that *Synechococcus* could play a major role in depleting the
Si of surface waters in this area, which are devoid of diatoms. During the OUTPACE cruise, there
were no clear correlations between *Synechococcus* distributions and the measured 0.4-3 µm BSi
concentrations. This could be explained by the extremely wide range of individual cellular Si
quotas estimated to vary between 1 and 4700 amol Si cell$^{-1}$ (with an average value of 43) from cells
collected in the North Western Atlantic (Ohnemus et al., 2016), where *Synechococcus* contributed
up to 23.5 % of $\Sigma BSi$ (Krause et al., 2017). In the latter study, a first-order estimate of the
contribution of *Synechococcus* to the global annual Si production flux amounted to 0.7-3.5%,
which is certainly low, but comparable to some other important input or output fluxes of Si (Tréguer
and De la Rocha, 2013).
**6 Conclusion**
The Sargasso Sea (BATS) and the North Tropical Pacific Ocean (ALOHA) were until now the
only two subtropical gyres where the Si cycle was fully investigated during time-series surveys. In
this paper, we provide the first complementary data from two cruises documenting production,
biomass and export fluxes from the oligotrophic to ultra-oligotrophic conditions in the South
Tropical Pacific Gyre, which may lower the estimates of diatom contribution to primary





productivity and export fluxes for the Pacific Ocean and for mid-ocean gyres in general. The mid-
ocean gyres (representing 1/3 of the global ocean) are severely under-sampled regarding the Si
cycle, and may encompass very different situations, in particular in the vicinity of Islands and
archipelagos with reduced bathymetry, and nutrient-fertilized surface waters, to HNLC waters and
even HNLSiLC along the equatorial divergence (Dugdale and Wilkerson, 1998). The mid-ocean
gyres contribution to Si production was recently revised down to 5-7% of the total by Brzezinski
et al. (2011) building on estimates from the North Subtropical Pacific Gyre. The present study
points to even lower values for the South Pacific Gyre, confirming its ultra-oligotrophic nature,
and should further decrease this estimate. These findings clearly warrant for improved coverage of
these areas and for more complete elemental studies (from Si production to export).
Diatom community structure and contribution to total biomass could be summarized by
differentiating 3 main ecosystems: (i) the eutrophic Peru-Chile coastal upwelling, where colonial
neritic centric diatoms such as *Skeletonema* sp., *Chaetoceros* sp. and *Thalassiosira* sp. contributed
to elevated abundances (>100,000 cells L$^{-1}$) and very high Si uptake rates; (ii) the HNLC region
off the Marquesas Islands and the nutrient depleted but Fe-fertilized region of the Melanesian
Archipelago, where a distinct community largely dominated by small and medium-sized pennates
such as *Cylindrotheca closterium* and *Pseudo-nitzschia delicatissima* developed to moderate levels
(<30,000 cells L$^{-1}$), while Fe levels in the MA further stimulated diazotrophs and DDAs which
could have stimulated diatom growth through secondary N release; (iii) the SPG, characterized by
ultra-oligotrophic conditions and Fe-limitation, where diatoms reached negligible abundances
(<200 cells L$^{-1}$) with species typical of oligotrophic regions, such as *Nitzschia bicapitata*,
*Mastogloia woodiana*, *Planktoniella sol* as well as radiolarians.
Finally, thanks to both size-fractionated biomass and Si uptake measurements, we were able to
confirm a potential role for *Synechococcus* cells in Si uptake in all environments, which may be of
importance relative to diatoms in oligotrophic regions, but probably negligible in highly productive
regions such as coastal upwellings. Mechanisms linked to Si uptake in *Synechococcus* and its
ecological function still need to be elucidated, and further attention to the Si cycle needs to be
placed on this elusive pico- and nano-sized fraction.




**7 Data availability**
**8 Author contribution**
KL treated all data and wrote the paper. BQ and PR sampled on board and analyzed Si data from
the BIOSOPE cruise. SH-N and O.G. collected nutrient samples on board and analyzed nutrient
data from the OUTPACE cruise. VC sampled for all BSi data and diatom diversity on board, and
analyzed plankton net samples on the OUTPACE cruise. CB analyzed all Si data and ran diatom
cell counts during her Masters thesis. HC and JR were in charge of all pigment data for both cruises.
NL collected and analyzed Si export flux data from the OUTPACE drifting sediment traps.
**9 Competing interests**
The authors declare that they have no conflict of interest.
**10 Special Issue Statement**
This article is part of the special issue "Interactions between planktonic organisms and
biogeochemical cycles across trophic and $N_2$ fixation gradients in the western tropical South Pacific
Ocean: a multidisciplinary approach (OUTPACE experiment)"
**11 Acknowledgments**
This work is part of the OUTPACE Experiment project (https://outpace.mio.univ-amu.fr/) funded
by the Agence Nationale de la Recherche (grant ANR-14-CE01-0007-01), the LEFE-CyBER
program (CNRS-INSU), the Institut de Recherche pour le Développement (IRD), the GOPS
program (IRD) and the CNES (BC T23, ZBC 4500048836), and the European FEDER Fund under
project 1166-39417). The OUTPACE cruise (http://dx.doi.org/10.17600/15000900) was managed
by the MIO (OSU Institut Pytheas, AMU) from Marseilles (France). The BIOSOPE project was
funded by the Centre National de la Recherche Scientifique (CNRS), the Institut des Sciences de
l'Univers (INSU), the Centre National d'Etudes Spatiales (CNES), the European Space Agency
(ESA), The National Aeronautics and Space Administration (NASA) and the Natural Sciences and
Engineering Research Council of Canada (NSERC). This is a contribution to the BIOSOPE project
of the LEFE-CYBER program. The project leading to this publication has received funding from



European FEDER Fund under project 1166-39417. We warmly thank the captain, crew and CTD
operators on board R/V l'Atalante during both cruises. We further acknowledge Fernando Gomez
for providing diatom cell counts during BIOSOPE, Dr Jeremy Young at University College of
London for allowing the use of Parmale image from the Nannotax website and Mathieu Caffin for
providing a NanoSIMS image of *Synechoccocus* collected during OUTPACE showing cellular Si
accumulation.

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





**13 Figure Legend**
**Figure 1:** Bathymetric map of the stations sampled in the South Pacific Ocean during the OUTPACE cruise (Feb.-Apr. 2015) and
the BIOSOPE cruise (Oct.-Nov. 2004). Short-term duration stations are indicated in white, and long-term duration stations (typically
2-3d) in black.
**Figure 2:** Nutrient distribution (orthosilicic acid, nitrate, phosphate, in μM) along the OUTPACE cruise transect.
**Figure 3:** Nutrient distribution (orthosilicic acid, nitrate, phosphate, in μM) along the BIOSOPE cruise transect.
**Figure 4 :** Top panel: TChl$a$ distribution during the OUTPACE cruise in the SW Pacific (in μg L$^{-1}$) with fucoxanthin overlay lines
in white (in ng L$^{-1}$). Lower panel: TChl$a$ distribution during the BIOSOPE cruise in the SW Pacific (in μg L$^{-1}$) with fucoxanthin
overlay lines in white (in ng L$^{-1}$). Black dots indicated the Ze depth.
**Figure 5:** a.c Biogenic silica (BSi) and b.d. Lithogenic Silica (LSi) distribution during the OUTPACE and BIOSOPE cruises
respectively (in μmol L$^{-1}$).
**Figure 6:** a.b Size-fractionated integrated Biogenic silica ($\Sigma$ BSi) standing stocks (0-125 m) during the BIOSOPE cruise. UPW1
stations was only integrated over 50 m and UPX1 and UPX2 over 100 m. The b panel shows a zoom over the central section where
integrated BSi stocks are an order of magnitude lower than at the two extremities of the transect. Grey bars indicate that no size-
fractionation was conducted and represent the total $\Sigma$ BSi. C. Size-fractionated integrated Biogenic silica ($\Sigma$ BSi) standing stocks
(0-125 m) during the OUTPACE cruise.
**Figure 7:** a. Total absolute Si uptake rates ($\rho$ Si) vertical profiles (in μmol L$^{-1}$ d$^{-1}$) at the LD stations MAR, HNL, GYR, EGY,
UPX and UPW. b. ρSi in the 0.2 - 2 μm size fraction at the same sites.
**Figure 8:** Si uptake kinetic experiments conducted at the LD stations MAR, HNL, GYR, EGY, UPX at various euphotic depths.
Specific Si uptake rates (in d$^{-1}$) are plotted vs Si(OH)$_4$ increasing concentrations. Data was adjusted with hyperbolic curves when
statistically relevant and V$_{max}$ and K$_S$ values indicated below each curve.
**Figure 9:** Diatoms cellular concentrations (cells L$^{-1}$) derived from a. Niskin cell counts, b. number of taxa and c. relative contribution
to POC biomass (%) at the surface and DCM levels during the OUTPACE cruise.
**Figure 10:** Diatoms cellular concentrations (cells L$^{-1}$) derived from Niskin cell counts at several depths during the BIOSOPE cruise
(data from Gomez et al. 2007).
**Figure 11:** Potential siliceous organisms in the picoplanktonic (<2-3 μm) size fraction. a.Siliceous scale-bearing Parmale
(*Tetraparma pelagica* in SEM, photo courtesy of Dr. J. Young), b. centric diatom (*Minidiscus trioculatus*), c. *Synechoccocus* cell
showing Si assimilation in red ($^{28}$Si) in NanoSIMS (photo courtesy of M. Caffin).








**14 Tables**
**Table 1: Size-fractionated integrated Si production rates in mmol Si m$^{-2}$ d$^{-1}$ in the SEP (BIOSOPE). Integrated Si production**
**was measured over the 0-1% light depth range for each site (in parenthesis in column 5), and normalized over 100 m**
**considering a zero production at 100 m in the last column.**

| Stations | ΣρSi <2μm | ΣρSi 2-10 μm | ΣρSi >10μm | Total ΣρSi | Total ΣρSi over 0-100 m |
|---|---|---|---|---|---|
| MAR1 | 0.15 | 0.51 | 4.37 | 5.02 (50 m) | 5.87 |
| HNL1 | 0.05 | 0.12 | 0.58 | 0.75 (80 m) | 0.77 |
| GYR2 | 0.01 | 0.01 | 0.02 | 0.04 (110 m) | 0.04 |
| EGY | 0.03 | 0.07 | 0.09 | 0.19 (100 m) | 0.19 |
| UPW2 | 0.62 | 2.88 | 39.66 | 43.16 (35 m) | 52.36 |
| UPX1 | 1.07 | 5.90 | 13.49 | 20.46 (30 m) | 42.46 |


**Table 2: Dominant diatom species in each main system of the BIOSOPE and OUTPACE cruises. Taxonomic information**
**for the OUTPACE cruise are derived from discrete samplings at the surface and DCM and phytoplankton nets, while**
**information for the BIOSOPE cruise were obtained through an average of six discrete samples over the euphotic layer (see**
**Gomez et al., 2007).**

| Cruise | Oceanic system | Dominant diatom species |
|---|---|---|
| OUTPACE | Melanesian Archipelago | *Pseudo-nitzschia spp. & Pseudo-nitzschia delicatissima, Cylindrotheca closterium, Mastogloia woodiana, Leptocylindrus mediterraneus, Hemiaulus membranaceus, Chaetoceros spp. (hyalochaete), Pseudosolenia calcar-avis, Climacodium frauenfeldianum, Planktoniella sol* |
| | South Pacific Gyre | *Climacodium frauenfeldianum, Pseudo-nitzschia spp., Chaetoceros spp. (hyalochaete), Pseudo-nitzschia delicatissima, Mastogloia woodiana* |
| BIOSOPE | Western HNLC area (Marquesas) | *Pseudo-nitzschia delicatissima, Rhizosolenia bergonii, Thalassiothrix longissima, Plagiotropis spp., Pseudo-nitzschia pungens, P. subpacifica* |
| | South Tropical Pacific | *Nitzschia bicapitata species complex, Nitzschia sp., Thalassiothrix longissima, Pseudo-nitzschia delicatissima* |
| | South Pacific Gyre | *Hemiaulus hauckii, Chaetoceros curvisetus, Bacteriastrum cf. comosum* |
| | Eastern Gyre | *Pseudo-nitzschia cf. delicatissima, Pseudo-nitzschia cf. subpacifica, Pseudoeunotia sp.* |
| | Peru-Chile Upwelling | *Chaetoceros compressus, Bacteriastrum sp., Thalassiosira subtilis, Chaetoceros cf. diadema, Skeletonema sp., Pseudo-nitzschia sp.* |







**Table 3: Particulate biogenic and lithogenic (BSi and LSi) Silica in drifting sediment traps at each long duration station**
**during OUTPACE cruise, at 153, 328 and 519 m depth.**

|   | Trap depth | BSi | LSi |
|---|---|---|---|
|   | m | µmol Si m$^{-2}$ d$^{-1}$ | µmol Si m$^{-2}$ d$^{-1}$ |
| **A** | 153 | 0.5 | 23.1 |
|   | 328 | 0.2 | 4.6 |
|   | 519 | 0.1 | 5.2 |
| **B** | 153 | 2.6 | 0.4 |
|   | 328 | 2.9 | 0.6 |
|   | 519 | 4.8 | 1.1 |
| **C** | 153 | 1.8 | 0.5 |
|   | 328 | 0.5 | 0.2 |

**Table 4: Integrated Si production rates in various systems for comparison with our study from direct $^{32}$Si uptake**
**measurements or from indirect silicate utilization (ΔSiO$_4$) estimates (*).**

| Region | Integrated Si production rate $\Sigma\rho Si$ (mmol m$^{-2}$ d$^{-1}$) | References |
|---|---|---|
| **Coastal upwellings** | | |
| BIOSOPE: Peru-Chile upwelling | **42 – 52 (UPW)** | *This study* |
| Baja California | 89 | Nelson and Goering, 1978 |
| Monterey Bay | 70 | Brzezinski et al., 1997 |
| Peru | 27 | Nelson et al., 1981 |
| Southern California Current coastal waters | 1.7 – 5.6 | Krause et al., 2015 |
| **Oceanic area** | | |
| BIOSOPE: South Eastern Pacific (HNLC) | **0.8 – 5.6 (HNL – MAR)** | *This study* |
| Gulf Stream warm rings | 6.4 | Brzezinski and Nelson, 1989 |
| Central Equatorial Pacific (HNLC) | 3.9 | Blain et al., 1997 |
| North Pacific (OSP) | 5.1 | Wong and Matear, 1999* |
| North Atlantic (POMME) | 1.7 | Leblanc et al., 2005b |
| North Atlantic (Bengal) | 0.9 | Ragueneau et al., 2000 |
| Mediterranean Sea (SOFI) | 0.8 | Leblanc et al., 2003 |
| **Oligotrophic area** | | |
| BIOSOPE: South Eastern Pacific Gyre | **0.04 (GYR) – 0.2 (EGY)** | *This study* |
| Central Equatorial Pacific | 0.8 – 2.1 | Blain et al., 1997 |
| Eastern Equatorial Pacific | 0.2 – 2.5 | Leynaert et al., 2001 ; Adjou et al., 2011 ; Krause et al., 2011, Demarest et al., 2011 |
| Central North Pacific | 0.5 – 2.9 | Brzezinski et al., 1998 |
| North Pacific Subtropical Gyre | 0.1 – 1.7 | Krause et al., 2013 |
| North Pacific Subtropical Gyre (ALOHA) | 0.1 – 0.5 | Brzezinski et al., 2011 |
| Sargasso Sea | 0.5 | Brzezinski and Nelson, 1995 |
| Sargasso Sea (BATS) | 0.1 – 0.9 | Brzezinski and Kosman, 1996 (1996), Nelson and Brzezinski, 1997 |




**Table 5: Summary of ΣBSi stocks in mmol Si m$^{-2}$ for the OUTPACE and BIOSOPE and other oceanic and oligotrophic systems.**











| Region | Average Integrated Si biomass ΣBSi (mmol m$^{-2}$) | References |
|---|---|---|
| **Coastal upwellings** | | |
| BIOSOPE: Peru-Chile upwelling | **65.7 ± 53.8** | *This study* |
| Southern California Current coastal waters | 53.2 ± 39.3 | Krause et al., 2015 |
| **Oceanic area** | | |
| Southern California Current oceanic waters | 1.6 ± 0.3 | Krause et al., 2015 |
| BIOSOPE: South Eastern Pacific (HNLC) | **11.9 ± 10.9** | *This study* |
| **Oligotrophic area** | | |
| Mediterranean Sea (BOUM) | 1.1 – 28.2 | Crombet et al., 2011 |
| Sargasso Sea (BATS) | 4.0 ± 6.8 | Nelson et al., 1995 |
| Sargasso Sea | 0.9 – 6.1 | Krause et al., 2017 |
| North Pacific Subtropical Gyre | 1.6 – 12.8 | Krause et al., 2013 |
| North Pacific Subtropical Gyre (ALOHA) | 3.0 ± 1.1 | Brzezinski et al., 2011 |
| Central North Pacific | 7.1 ± 3.0 | Brzezinski et al., 1998 |
| Eastern Equatorial Pacific | 3.8 – 18.0 | Krause et al., 2011 |
| BIOSOPE: South Eastern Pacific Gyre | **1.1 ± 1.1** | *This study* |
| OUTPACE: South Western Pacific Gyre | **1.0 ± 0.2** | *This study* |
| OUTPACE: Melanesian Archipalago | **2.4 ± 1.0** | *This study* |






**Table 6: Summary of Si export fluxes in sediment traps at various depths in μmol Si m$^{-2}$ d$^{-1}$ for the OUTPACE cruise compared to other studies.**

| Region | Sediment trap depth (m) | Average Si export fluxes (μmol m$^{-2}$ d$^{-1}$) | References |
|---|---|---|---|
| **Coastal upwellings** | | | |
| Southern California Current coastal waters | 100 | 8,000 ± 5,760 | Krause et al., 2015 |
| **Oceanic area** | | | |
| North Atlantic (NABE) | 400 | 10 – 145 | Honjo and Manganini, 1993 |
| North Atlantic (POMME) | 400 | 2 - 316 | Mosseri et al., 2005 ; Leblanc et al., 2005b |
| North Pacific Subtropical Gyre (ALOHA) | 150 | 14 - 300 | Brzezinski et al., 2011 |
| **Oligotrophic area** | | | |
| Sargasso Sea (BATS) | 150 | 17 - 700 | Nelson et al., 1995 |
| Sargasso Sea (BATS) | 150 | 130 | Brzezinski and Nelson, 1995 |
| | 200 | 113 | |
| | 300 | 85 | |
| OUTPACE: South Western Pacific Gyre | 153 | **1.8** | *This study* |
| | 328 | **0.5** | |
| OUTPACE: Melanesian Archipelago | 153 | **1.6** | *This study* |
| | 328 | **1.6** | |
| | 519 | **2.5** | |







**15 Appendices**

| Stations | ΣBSi 0.2-2 μm (mmol m⁻²) | ΣBSi 2-10 μm (mmol m⁻²) | ΣBSi >10 μm (mmol m⁻²) | Total ΣBSi (mmol m⁻²) |
|---|---|---|---|---|
| MAR1 | 0.36 | 3.49 | 20.28 | 24.12 |
| NUK1 | 0.34 | 0.66 | 2.40 | 3.40 |
| HNL1 | 0.20 | 2.34 | 5.54 | 8.09 |
| 1 | | | | 3.79 |
| 2 | | | | 0.40 |
| 3 | | | | 0.48 |
| 4 | | | | 0.31 |
| 5 | | | | 0.20 |
| 6 | | | | 0.18 |
| 7 | | | | 0.20 |
| 8 | | | | 0.49 |
| GYR2 | 0.30 | 0.37 | 0.55 | 1.23 |
| GYR5 | 0.13 | 0.24 | 0.39 | 0.75 |
| 11 | | | | 0.42 |
| 12 | | | | 0.82 |
| 13 | | | | 0.16 |
| 14 | | | | 0.47 |
| 15 | | | | 1.03 |
| EGY2 | 0.29 | 0.45 | 0.87 | 1.60 |
| EGY4 | 0.15 | 0.25 | 0.65 | 1.05 |
| 17 | | | | 2.36 |
| 18 | | | | 2.47 |
| 19 | | | | 0.45 |
| 20 | | | | 1.50 |
| 21 | | | | 3.48 |
| UPW1* | 1.27 | 5.36 | 55.43 | 62.05 |
| UPW2 | 3.75 | 15.28 | 124.10 | 142.81 |
| UPX1** | 7.66 | 9.80 | 14.64 | 32.00 |
| UPX2** | 2.27 | 8.12 | 15.49 | 25.88 |


**Appendix 1: Integrated size-fractionated Biogenic Silica concentrations (ΣBSi) in the South Eastern Pacific (BIOSOPE**
**cruise) over 0-125 m. 0-50 m for * and 0-100 m for **.**



| Stations | ΣBSi 0.4-3 μm (mmol m$^{-2}$) | ΣBSi > 3 μm (mmol m$^{-2}$) | Total ΣBSi (mmol m$^{-2}$) |
|---|---|---|---|
| 1 | 1.24 | 2.52 | 3.76 |
| 2 | 0.39 | 3.56 | 3.95 |
| 3 | 0.43 | 1.83 | 2.26 |
| A | 0.26 | 1.83 | 2.09 |
| 4 | 1.06 | 2.24 | 3.30 |
| 5 | 0.51 | 3.60 | 4.11 |
| 6 | 0.70 | 1.80 | 2.49 |
| 7 | 0.39 | 1.95 | 2.34 |
| 8 | 0.39 | 1.12 | 1.51 |
| 9 | 0.50 | 1.45 | 1.96 |
| 10 | 0.77 | 0.98 | 1.75 |
| 11 | 0.24 | 1.00 | 1.24 |
| 12 | 0.17 | 1.29 | 1.46 |
| B | 0.30 | 1.60 | 1.89 |
| 13 | 0.17 | 0.96 | 1.13 |
| C* | 0.50 | 0.93 | 1.43 |
| C* | 0.59 | 1.03 | 1.61 |
| 14* | 0.68 | 1.02 | 1.70 |
| 15* | 0.76 | 1.38 | 2.14 |


**Appendix 2: Integrated size-fractionated Biogenic Silica concentrations (ΣBSi) in the South Western Pacific (OUTPACE**
**cruise) over 0-125 m and 0-200 m for \*.**






| STATION | 1 | 2 | 3 | A | A | A | A | A | 4 | 5 | 6 | 7 | 8 | 9 | 10 | 11 | 12 | B | B | B | B | B | C | C | C | C | C | 14 | 15 |
|---|---|---|---|---|---|---|---|---|---|---|---|---|---|---|---|---|---|---|---|---|---|---|---|---|---|---|---|---|---|
| Date | 22/02 | 23/02 | 24/02 | 26/02 | 27/02 | 28/02 | 1/3 | 2/3 | 4/3 | 5/3 | 6/3 | 7/3 | 8/3 | 9/3 | 10/3 | 11/3 | 12/3 | 15/3 | 16/3 | 17/3 | 18/3 | 19/3 | 23/3 | 24/3 | 25/3 | 26/3 | 27/3 | 29/3 | 30/3 |

**Diatoms**
- *Asterolampra marylandica*
- *Asteromphalus heptactis/roperianus*
- *Bacillaria paxillifera*
- *Bacteriastrum comosum*
- *Bacteriastrum elongatum*
- *Cerataulina cf pelagica*
- *Chaetoceros hyalochaetae spp/*
- *Chaetoceros compressus with Richelia*
- *Chaetoceros dadayi*
- *Chaetoceros peruvianus*
- *Climacodium frauenfeldianum*
- *Cylindrotheca closterium*
- *Dactyliosolen blavyanus*
- *Dactyliosolen fragilissimus*
- *Dactyliosolen phuketensis*
- *Ditylum brightwelli*
- *Gossleriella tropica*
- *Guinardia cylindrus with Richelia*
- *Guinardia striata*
- *Haslea sp.*
- *Helicotheca tamesis*
- *Hemiaulus membranaceus*
- *Hemiaulus hauckii*
- *Hemidiscus sp.*
- *Leptocylindrus mediterraneus*
- *Lioloma pacificum*
- *Navicula/Nitzschia/Mastogloia*
- *Nitzschia longissima*
- *Planktoniella sol*
- *Proboscia alata*
- *Pseudoguinardia recta*
- *Pseudolenia calcar-avis*
- *Pseudo-nitzschia*
- *Rhizosolenia sp. with Richelia*
- *Rhizosolenia imbricata/bergonii*
- *Rhizosolenia formosa*
- *Skeletonema sp.*
- *Stephanopyxis sp.*
- *Thalassionema sp.*
- *Triceratium sp.*
- *Undetermined pennates < 50 µm*
- *Undetermined pennates 100-200 µm*
- *Undetermined pennates >200 µm*
- *Thalassiosira-like ~15 µm*
- *Thalassiosira-like ~50 µm*
- *Thalassiosira-like ~100 µm*

**Radiolarians**
- *Single radiolarians*
- *Colonial radiolarians*

**Silicoflagellates**
- *Dictyocha speculum*

**Diazotrophs**
- *Trichodesmium spp.*
- *Richelia intracellularis*
- *Croccosphera sp.*
- *Other filamenteous cyanobacteria*


**Appendix 3: Semi-quantitative contribution of siliceous plankton (diatoms, radiolarians, silicoflagellates) and diazotrophs in plankton nets hauls of 35 µm mesh size (over 0-150 m at all sites except but over 0-200 m at stations 14 and 15) during the OUTPACE cruise. Long duration stations were sampled every day. Light grey, medium grey and dark grey correspond to minor, common and dominant abundances respectively.**

961

962



Figure 1

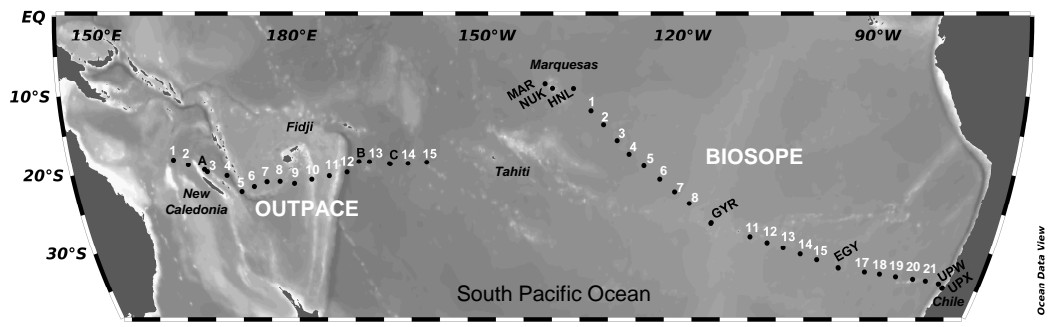



Figure 2

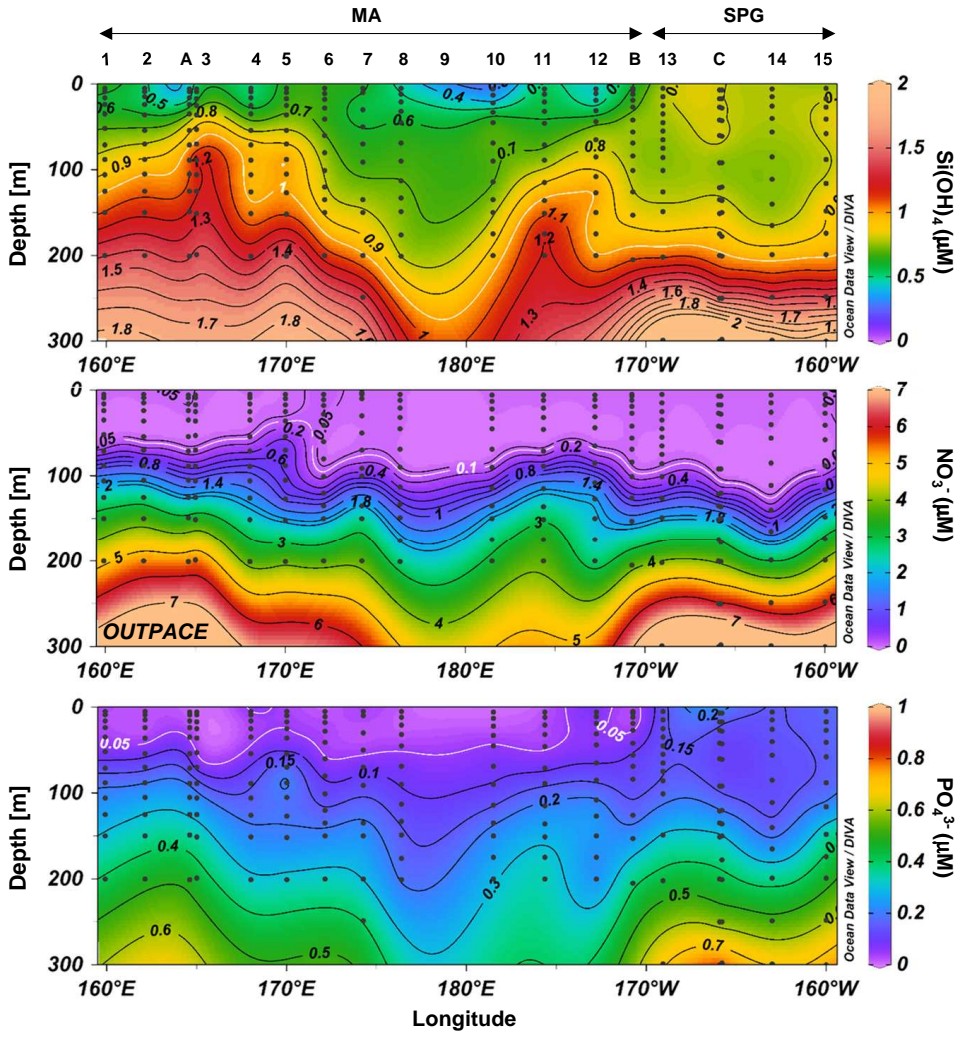



Figure 3

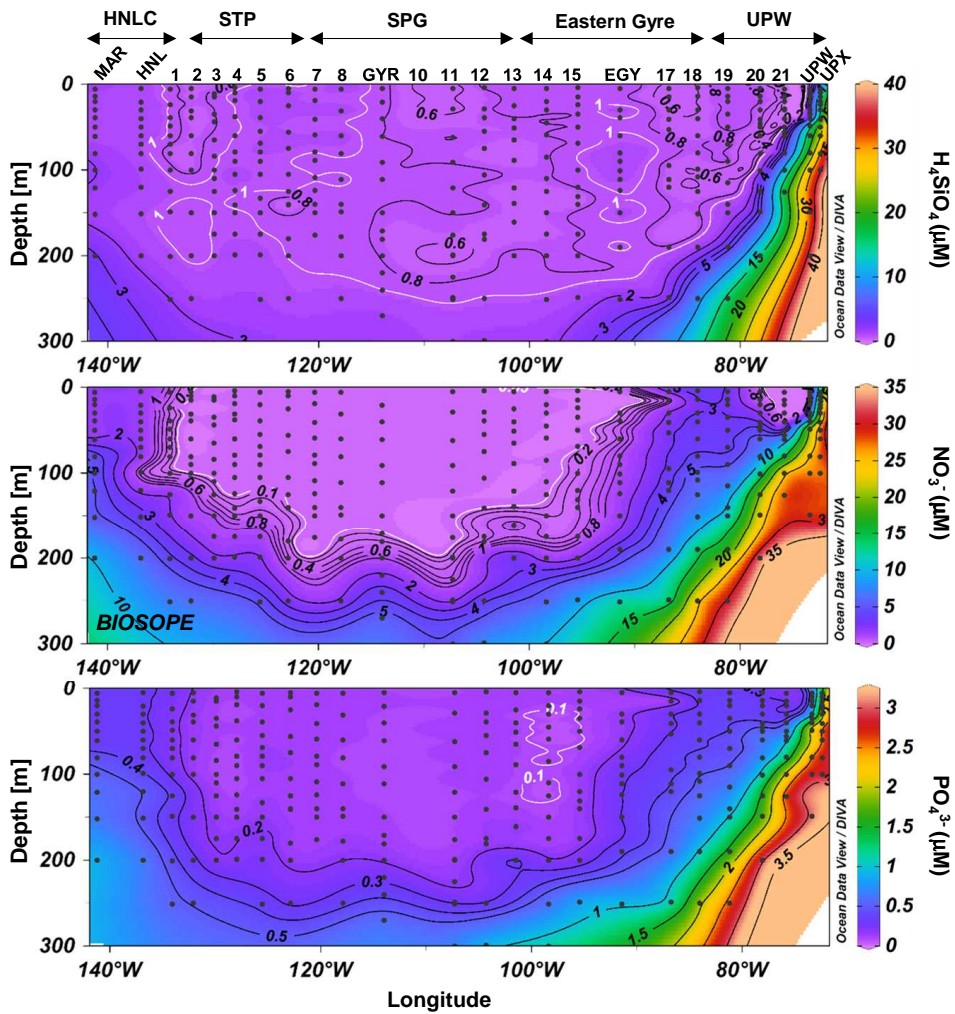



Figure 4

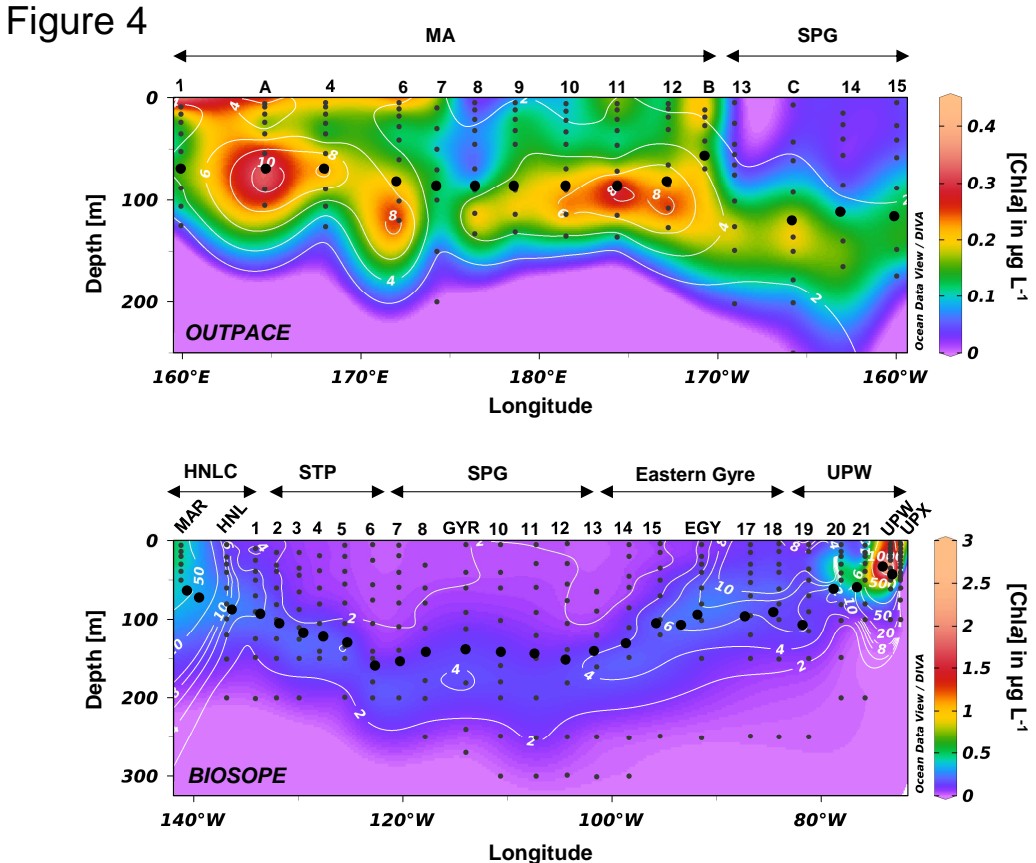



Figure 5

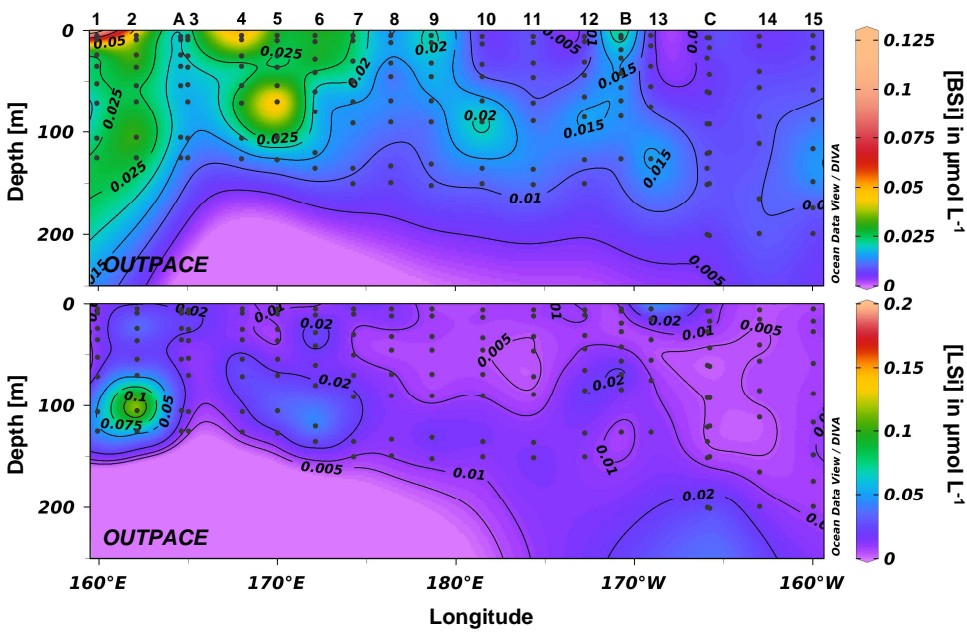

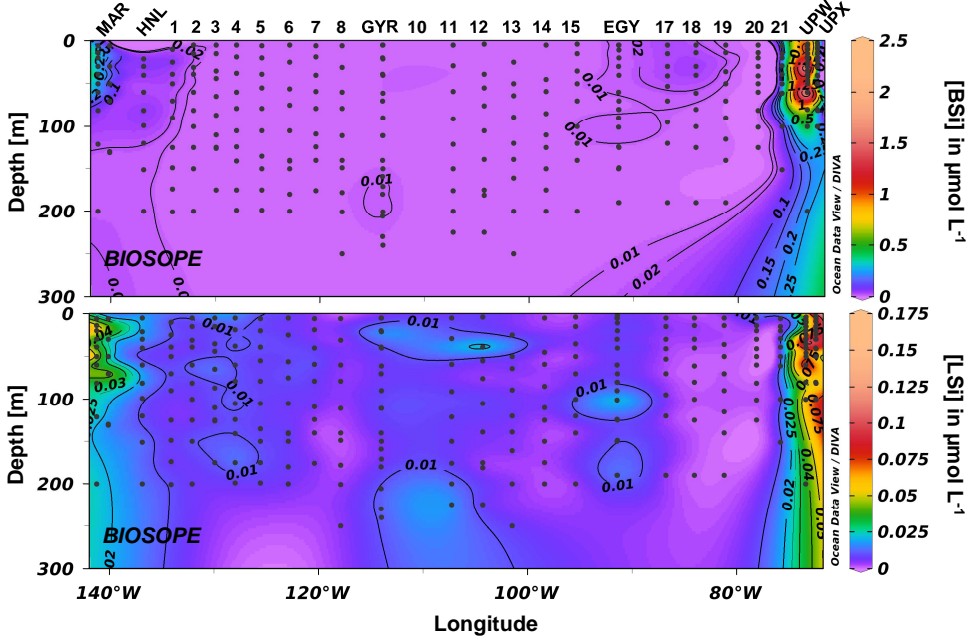



Figure 6

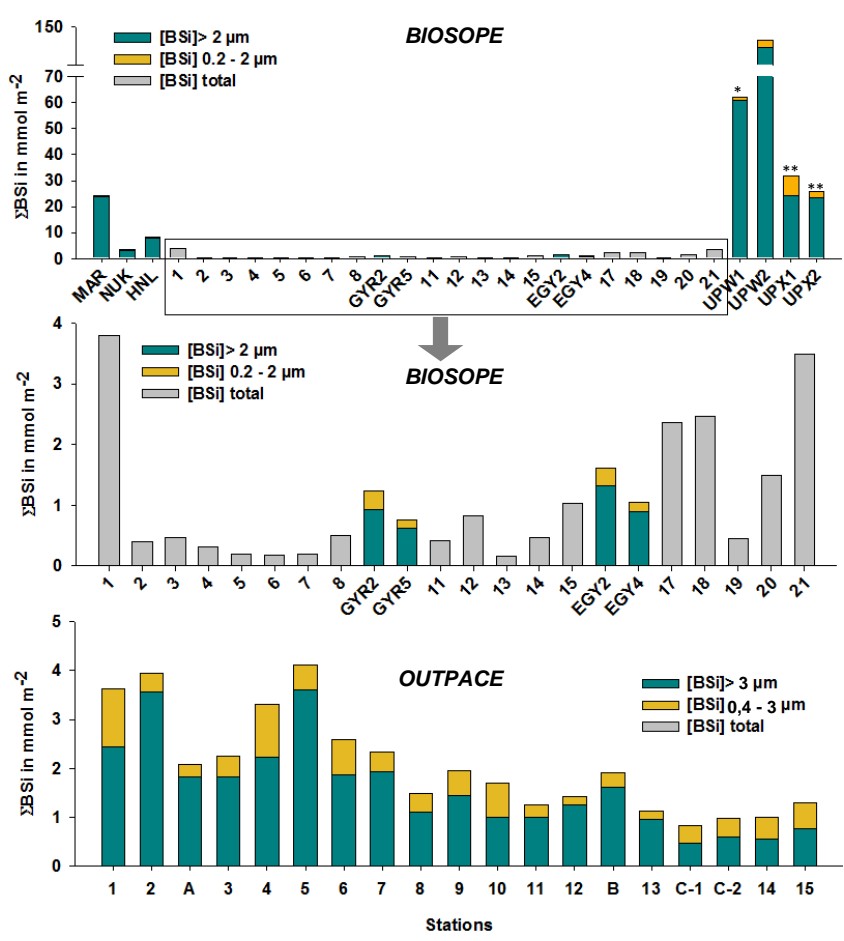





Figure 7

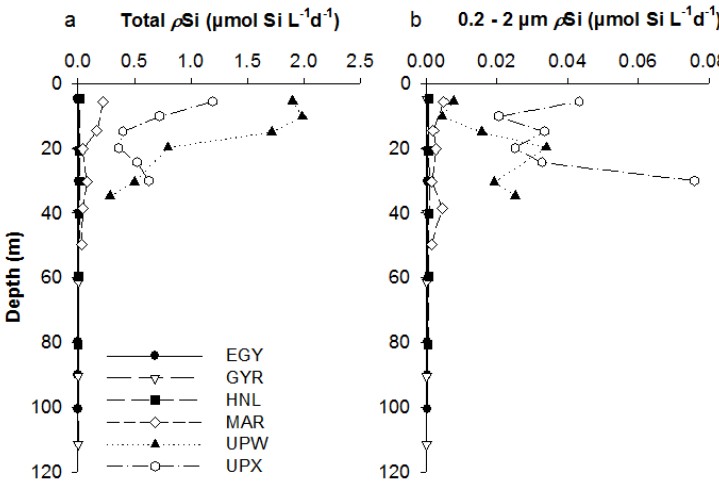



Figure 8

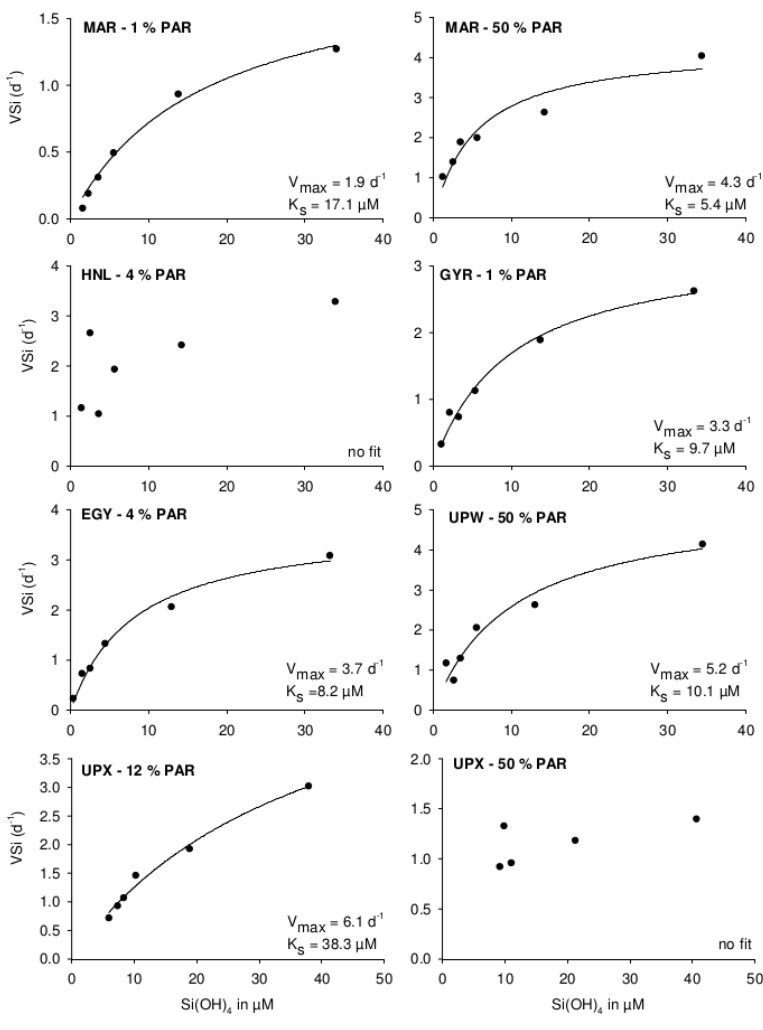



Figure 9

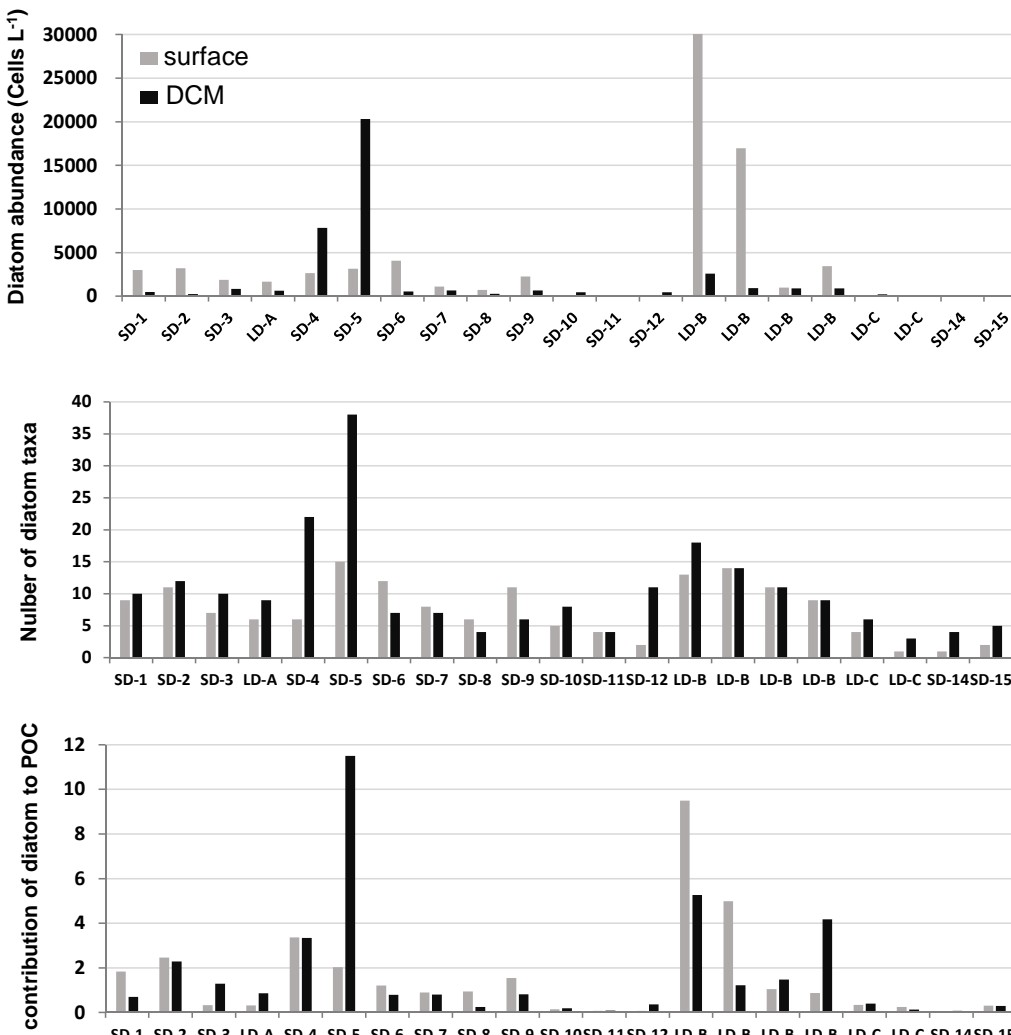



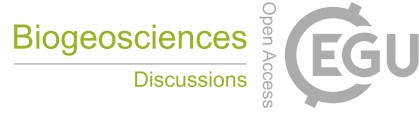

Figure 10

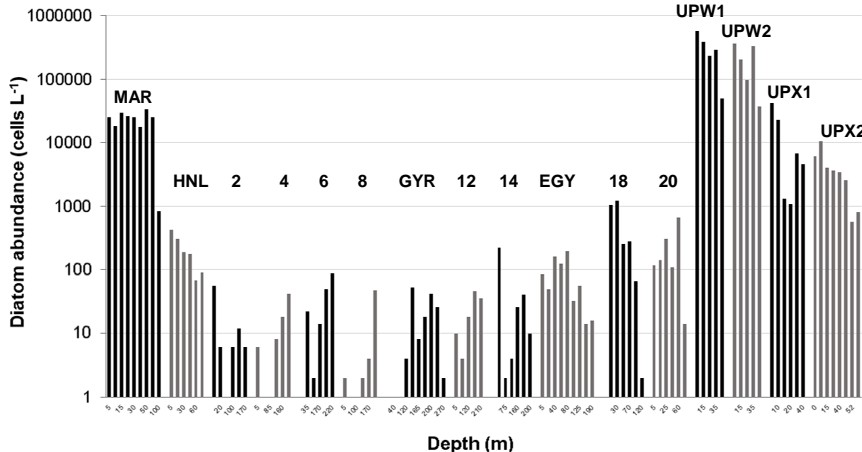



Figure 11

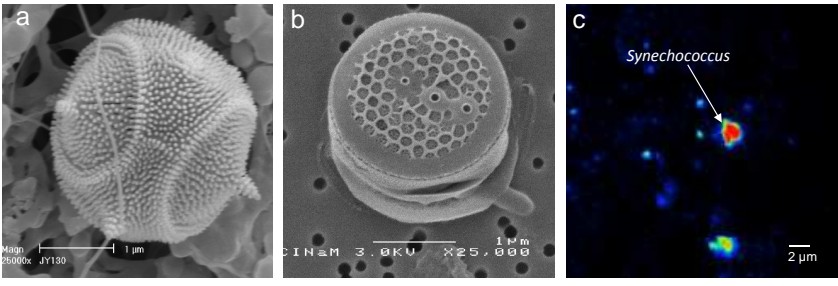