# Peer review of "Silicon cycle in the Tropical South Pacific: contribution to the global Si cycle and evidence for an active pico-sized siliceous plankton"

_Biogeosciences, 2018_

## Referee Comment (RC1) · Anonymous Referee #1 · 18 May 2018

General comments:

This is a very important paper for the marine silicon biogeochemical community. The data presented on the silica cycle in the ultra oligrotrophic South Pacific are the very first from this region and thus extremely valuable. While the extremely low biomass and silica production rates are not surprising it is extremely important that they be quantified. Those data aid in our understanding of the contribution of subtropical gyres to global silica standing stocks and silica production. While I am 100% in favor of seeing this data published it was disappointing that silica production rates were only measured at two truly oligotrophic stations. So while the authors use these data to place the

observed rates in a global context the extrapolation is extreme as fully acknowledged by the authors.

The authors were able to conduct some very fine kinetic studies that show active silicic acid uptake by the $< 2 \mu$m size fraction. Few diatoms would be expected in this size fraction pointing to uptake by non-diatoms. There is significant confusion as to the kinetic experiments in terms of size fractionation that must be clarified before publication. More on that below. The quantitation of diatom taxa and abundance is extensive and valuable.

I have no major issue with the interpretation of the data or the analyses. My suggestions for improvements are detailed below.

Specific comments: The title of the work emphasizes the finding that a significant fraction of the observed uptake was in the picoplankton size class. The paper contains so much more than this. Please consider expanding the title to something like "Silicon cycle in the Tropical South Pacific: contribution to the global Si cycle and evidence for an active pico-size siliceous plankton".

Line 40: This paragraph is very long. Maybe break it at line 40.

Line 42: The data available from the north Pacific subtropical gyre cited later in the paper would be relevant here as well.

Line 40-56. This is a suggestion only. Our understanding of the role of subtropical gyres in the global Si cycle began in the Sargasso Sea which through extrapolation led the fairly high estimates for the contribution of these regions to global silica production. Data from the north Pacific led to a reduction in that estimate and the data presented here from the south Pacific lower it yet again. So what we are learning is that the Pacific is very different form the Atlantic and that the North and South Pacific differ from each other. This perspective is lacking in this paragraph which focuses on extrapolating silica production to carbon. It might be worthwhile to add a section that stays focused

on silicon as later in the paper silicon budgets are presented.

Line 58: maybe 'studies provide evidence for a role...' rather than 'studies have furthermore evidenced a role'.

Line 75: Maybe '...strategies and analyses were conducted on both cruises..." rather than "...strategies and homogenous analyses were conducted ...".

Line85, 86: Maybe "... transects that employed a common sampling strategy of short and long duration stations." Rather than '...transects with similar sampling strategy of short and long duration stations."

Line 99: Given the very low nutrient concentrations it the reader would benefit from knowing the detection limits of the specific nutrient analyses employed.

Line 119: 'quarters' instead of '4'. 'Plastic' petri dishes right?

Line 127: What method was used to remove the interference from HF in the LSi colorimetric analysis: boric acid or dilution?

Line 131: Kinetic assays? Do you mean you conducted time courses to test the efficiency of different digestion times?

Line 138-139: Please elaborate. It is unclear how the addition of Si was used to correct for dissolution in the face of the combined influence of dissolution of captured siliceous particles and the admixture of ambient water.

Line 141 Si & VSi rather than Si/VSi. Si/VSi looks like you are dividing one rate by the other. Line 150: 'averaged' instead of 'average'

Line 151. Many details are missing from this section of the methods. There is no indication of size fractions. Later in the paper it is claimed that kinetics were size fractioned like biomass, but I only see one set of kinetic curves and it is not clear what size fraction they represent (Fig 8.). Also in this section there is no mention of a 32Si addition.

[Figure]

Lines 187-196. The observation that the nitracline is much deeper than the silicicline is also observed in the Sargasso but not nearly to the same extent. It might be interesting to speculate as to why these depths differ in the discussion.

Line 198: rather than 'The distribution of orthosilicic acid concentrations were less clearly contrasted, . . ." maybe 'Horizontal gradients were not as strong for orthosilicic acid. . ..".

Line 211: 'existed" rather than 'subsisted"

Line 212: "magnitude" instead of "amplitude".

Line 216: Maybe:" The Chla a distribution during BIOSCOPE was similar to that observed during OUPACE with extremely. . ."

Line 238: The units in the figures for BSi and LSi are in micromoles per liter whereas in the text the concentrations are discussed as nanomoles per liter. Be consistent. I would suggest changing the figure to nanomoles per liter as it gets rid of leading zeros.

2467: Maybe "LSi concentration was highest at both ends of the transect but concentrations remained below those of BSI with average LSi values. . ."

Line 271: Here the reader learns that kinetic experiments were size fractionated. Move this information to the Methods. More importantly only one size fraction is shown in fig 8. Where is the data from the other fraction? Also the legend for Fig 8 should indicate the fraction shown.

Line 273; Maybe ". . . rank order of most productive stations follow the pattern observed for BSi with the highest values observed at UPW followed by UPX and MAR stations."

Line 281: It might be useful to readers if the specific rates are also translated into implied doubling times as this will give a sense of how fast or slow growth might be in the various areas.

Line 295: It is unclear what size fraction is shown in the Fig 8. Fix legend. Also where

is the data for the other fraction. Please clarify.

Line 316: Maybe "the lowest" rather than "record low".

Line 359-360: Maybe "We obtained size-fractionated biomass and . . . OUTPACE program and size fractionated production. . .during the BIOSCOPE program.

Line 362: This is a long paragraph. Maybe break here.

Line 377: "documented" instead of "evidenced"

Line 387: Here is a place where the influence of data from the Pacific on global budgets can be emphasized. The contribution fell when data from the NPSG was added and now it goes down again when the south Pacific is considered.

Line 390: The limited number of measurements is disappointing, but treated objectively.

Line 408: The flux is indeed incredibly low: wow! However, my appreciation of this is vague given that I do not understand the correction for dissolution in the traps discussed above.

Line 426: Maybe: DCM's are common in mid-ocean gyres and are known to be often dominated by pico-sized phytoplankton (Chavez et al. 1996), Studies documenting. . . . . ."

Line 448: As I read this discussion I find the text informative but I wonder if the stated trends might be reinforced through a non-dimensional scaling or other analysis that would provide an objective way to illustrate many of the inferred trends.

Line 490: This is a very long paragraph. Maybe subdivide.

Line 542: Somewhere in this section the differences between the shape of the kinetic curves obtained here for pico-size fraction and those for cultured Synechococcus should be discussed. In culture Synechococcus have linear uptake kinetics within the concentration range examined here whereas the data from the South Pacific clearly

show a hyperbolic response. It's difficult to know for sure, but it might be possible that the organism responsible for Si uptake in the small size fraction in the South Pacific is something other than Synechococcus which would be very interesting.

Line 545-546: Confusing sentence. Maybe "Significant BSi in the pico-sized fraction could be explained as an artifact from detritus or the contribution from a previously unrecognized taxa."

Line 552 "by" rather than 'with"

Line 555: To finish this argument the expected shape of a curve resulting purely from fragments should be given. I would think the signal would then be very noisy and inconsistent which is not observed.

---

## Referee Comment (RC2) · Anonymous Referee #2 · 21 May 2018

This is a good paper with timely and relevant information on a poorly studied region of the ocean. It has a both rate/biomass information along with floristic data, a combination not often available. It is unfortunate that the gyres had only a limited sampling. The data is tantalizing in what is seen, but more sampling in this area is required to confirm the extremely low rates. It is a great deal of information to present and there are some areas where either the paper structure or text is confusing. As I note below, the methods need considerable improvement. The description of replication and error bars is unacceptably vague. Claims of differences are not justified by any statistical analysis. There are very few measures of variability given and reader is left to wonder if replicates were even collected. Each measurement should have a standard

deviation, confidence interval or some other metric of variability. The methods need to explicitly note which samples were collected in duplicate, triplicate, etc. On line 150, the uptake measurements were noted to have a precision of 10-25% for the less productive station. What is it for more productive stations? To me, the use of separate figures for hydrology, nutrients, BSi, and rates creates difficulty in interpreting the information. Multiple pages of figures are needed to understand one cruise. It would be much more clear if all the data were in a single (or perhaps 2 adjacent) multi-panel figures. However, it requires a rewrite of the manuscript to discuss each cruise in parallel rather than dealing with hydrology, nutrients, etc from the two cruises together. Since the cruises are very separate in time and space, there is no reason to treat one data type at a time. Cell counts are very time consuming and tedious. Thus, it is always disappointing when the information is lumped into a single pool of diatoms in Figures. From the methods, it is quite impossible to determine if diatom counts were from the same depths as the BSi or a subset. Please clarify this. If the data density is there, please add this to the figures as a contour plot. The data availability statement is not present nor is there an explanation of why it is not present. This is not acceptable and I cannot recommend publication until this condition is met (as noted in the Instructions to Authors for the journal). The figures lack panel labels except for Fig. 7. This needs to be corrected for publication. Paragraph breaks need to be used for clarity, be they line spacing or indentations.

Line: comment 19: Chlorophyll does not need to be capitalized. 33-34: I am not sure what "silica production. . .comparable to . . .all areas of diatomaceous sediment" means. One is a rate per volume per time, the other is mass per volume sediment. Please clarify. 39: need to define chl a abbreviation first. 50-56: While these cited authors suggested these mechanisms may be leading to diatoms blooms, they have little direct experimental or observational data to this point. Wilson (2011) was later modified when a stratification value was discovered to be to high (see later work by Wilson et al. 2013) and Calil et al and Guidi et al. have done much more direct work on the role of mesoscale features than Krause et al. (2009, 2010). These are all key points to make,

but please cite the correct papers. 104: please provide temperature and length of pre-combustion 116: cascading is probably not the best word choice. Sequential or stacked is more accurate. 123: digestion, not attack 122-134. I am curious how standards were treated to have the same pH value as the samples. Si is a pH sensitive assay, so this merits some consideration. 143: please specify how the light measurement was made and then applied to generate the incubation depth. 151: Si uptake from the chlorophyll maximum. This description needs clarification. Was uptake measured as per section 3.7 or were changes in BSi measured as per section 3.6? The kinetic curve incubation lasted 8 hours, the in-situ incubation lasted dawn to dusk. Are there potential artifacts associated with the timing of division cycles? Later in the paper, it appears isotope uptake experiments were conducted, but the reader should not have to wait until then to know this. Finally, how relevant is this measurement to the waters above the DCM? 162: please list the net specifics: mouth opening and mesh size 216: This sentence is not clear. Please rephrase. It is apparently a comparison joined by the word than. I'm not sure what you are trying to say. 266: attributed, not assimilated. 268: What is this unit of variability? Standard deviations? confidence interval? If you wish to say they are different, please refer to a statistical test showing this. The $\pm$ ranges overlap considerably. I am not convinced. 269+: The same comment applies here. Are the duplicates? Triplicates? Error bars? Statistics? The rates have up to 25% precision errors, so this is important. 312+: contribution to biomass implies some conversion to a common currency (carbon, chl). How did you do this? 322: richness based on quantitative counts or the net tows? In either case, the authors need to specifiy the total number of cells examined. If it is 50 cells in one case and 500 in another, that will clearly influence the community richness observed. 326: Dominance within the diatom community needs to be specified as based on abundance or size/surface area. One large Coscinodiscus or Rhizosolenia will equal many small bicapitate Nitzschia. The Table 1 citation seems out of place. I think you mean Table 2. 489: The authors may wish to consider the work of Shipe et al. (1999) where they noted large rare diatoms contributed up to 26% of the Si uptake in the north Pacific. There is no information

on these giant diatoms, either solitary or aggregated, from the south Pacific. Any observations they have on this would be very relevant. 593: This study is not a time series as per HOT and BATS, so the topic sentence implication that this work adds to time-series work in the south Pacific is not correct.

Figures Fig. 4: The change in color scale is a bit confusing since the tendency to compare the two transects. If Fig. 4 Outpace were the same color scale as the Bioscope figure, then all the detail of the DCM would disappear. Likewise, the use of the Outpace color scale for the Bioscope would create detail.

Fig. 9: there are typos in the 2nd panel figure axis.

---

## Referee Comment (RC3) · Anonymous Referee #3 · 6 Jun 2018

Overall this is a solid study which presents a wealth of data from a vast and undersampled region. While not groundbreaking, it could be impactful if it spurs more study of Si cycling in this region. Generally, I agree with most of the study (the authors have done a commendable job with the cell count and taxonomy components) but have a few main comments:

- The contribution of Synechococcus: the authors have compelling data which is consistent with recent studies but this facet is under developed. Given the Silicon per cell for Synechococcus in the two publications these authors cite (Baines et al. 2012, Ohnemus et al. 2016), could they do a similar budget of Synechoccocus silica here?

[Figure]

Given the size of this project, surely there must be some flow cytometry data.

- Additionally, the isotope data is excellent to have but the rates for the kinetic data are unrealistic and not adequately discussed. For instance, 3.0 d-1 implies 4.3 doublings per day, 4.0 d-1 implies 5.8 doublings per day. Among all the experiments shown in Fig 8, all rates are exceptionally high as to be not believable. I think the authors need to better justify whether these data are useful and, if not, then perhaps consider eliminating.

Beyond these issues, I have numerous minor comments:

Line 64-66: Baines's estimates were indirect and extrapolated significantly, and were based on bSi associated with living cells (instead of total bSi).

Line 118: why the difference in filter sizes? Does this affect your results and interpretations?

Line 128: given such low bSi measured, it seems like this precision is quite high (i.e. high noise to signal ratio). May the authors please explain why they would not consider this an issue?

Line 149: Cerenkov counting is much less efficient than standard liquid scintillation methods correct? Given the low biomass (and thus low sample signal), did the Cerenkov background counts allow adequate resolution of analytically significant signals?

Line 154: why go up to 36 uM? Are there prior studies which have gone this high? Recent work (Shrestha & Hildebrand 2015) show that above 25 uM diatoms start turning off silicon transporters.

Line 229: given the high values, would the median (instead of average) be better here?

Line 275: 15 nmol/L/d given such low bSi means these cells are pretty active (e.g. 1 doubling per day)

Line 281, 298-299: Vmax is so high, it seems to be an error (see general comment).

Line 296: it doesn't say in the figure caption that these are just for pico sizes, please clarify.

Line 353: what is the percent dissolution among these samples, could those be used to infer dissolution rates in the water column and compare to biomass-specific rates?

Line 564: may you cite evidence for siliceous parmales in this region, aren't these only routinely observed in the subarctic North Pacific.

Line 582: how so? There are two problems: the quotas published by Ohnemus et al. 2016 are low and the standing stock of picoplankton isn't high enough to consistently drawdown Si. Second, if these standing stocks did get high enough, then to remove Si, this material would need to be exported; yet the export rates quantified in this region were the lowest observed. This feels like a disconnect.

Figure 2, 3: could the color scale be more logarithmic (like in Figure 4) and similar to allow easier comparison?

Figure 7: perhaps a log scale to see the low values easier?

Figure 9: please detail how the lower panel values were calculated

---

## Author Comment (AC1) · 9 Jul 2018

**General comments:**

This is a very important paper for the marine silicon biogeochemical community. The data presented on the silica cycle in the ultra oligrotrophic South Pacific are the very first from this region and thus extremely valuable. While the extremely low biomass and silica production rates are not surprising it is extremely important that they be quantified. Those data aid in our understanding of the contribution of subtropical gyres to global silica standing stocks and silica production. While I am 100% in favor of seeing this data published it was disappointing that silica production rates were only measured at two truly oligotrophic stations. So while the authors use these data to place the observed rates in a global context the extrapolation is extreme as fully acknowledged by the authors. The authors were able to conduct some very fine kinetic studies that show active silicic acid uptake by the < 2 μm size fraction. Few diatoms would be expected in this size fraction pointing to uptake by non-diatoms. There is significant confusion as to the kinetic experiments in terms of size fractionation that must be clarified before publication. More on that below. The quantitation of diatom taxa and abundance is extensive and valuable. I have no major issue with the interpretation of the data or the analyses. My suggestions for improvements are detailed below.

**Specific comments**: The title of the work emphasizes the finding that a significant fraction of the observed uptake was in the picoplankton size class. The paper contains so much more than this. Please consider expanding the title to something like "Silicon cycle in the Tropical South Pacific: contribution to the global Si cycle and evidence for an active pico-size siliceous plankton".

Thank you for your suggestion, we have corrected the title accordingly.

Line 40: This paragraph is very long. Maybe break it at line 40.
Corrected.

Line 42: The data available from the north Pacific subtropical gyre cited later in the paper would be relevant here as well.

Line 40-56. This is a suggestion only. Our understanding of the role of subtropical gyres in the global Si cycle began in the Sargasso Sea which through extrapolation led the fairly high estimates for the contribution of these regions to global silica production. Data from the north Pacific led to a reduction in that estimate and the data presented here from the south Pacific lower it yet again. So what we are learning is that the Pacific is very different form the Atlantic and that the North and South Pacific differ from each other. This perspective is lacking in this paragraph which focuses on extrapolating silica production to carbon. It might be worthwhile to add a section that stays focused on silicon as later in the paper silicon budgets are presented.

Indeed, we agree with both previous comments, and see how this persective is lacking.

We have rewritten that part of the introduction section as follows :

"Diatoms are known to contribute more importantly to primary production in meso- to eutrophic systems, yet several studies have emphasized that even if they are not dominant in oligotrophic regions, they may still contribute up to 10-20 % of C primary production in the Equatorial Pacific (Blain et al., 1997). In the oligotrophic Sargasso Sea (BATS station), their contribution was estimated to be as high as 26-48 % of new annual primary production (Brzezinski and Nelson, 1995) and to represent up to 30 % of Particulate Organic Carbon (POC) export, leading to an upward revision of the contribution of oligotrophic gyres to global Si budgets (Nelson and Brzezinski, 1997). Similar studies carried out in the Northern Pacific (HOT station) led to new estimates, as diatoms were found to be less important contributors to primary production. A combination of both Atlantic and North Pacific oligotrophic gyres budgets led to a revised contribution of 13 Tmol Si y$^{-1}$, a 51 % diminition of the previous estimate (Brzezinski et al., 2011)."

And in the conclusion:
"The mid-ocean gyres contribution to Si production was recently revised down to 5-7% of the total by Brzezinski et al. (2011) building on estimates from the North Subtropical Pacific Gyre. The present study points to even lower values for the South Pacific Gyre, confirming its ultra-oligotrophic nature, and should further decrease this estimate. These findings underscore the differences in functionning of different subtropical oligotrophic gyres between the North Atlantic, North Pacific and South Pacific and clearly warrant for improved coverage of these areas and for more complete elemental studies (from Si production to export).

Line 58: maybe 'studies provide evidence for a role. . .' rather than 'studies have furthermore evidenced a role'.
Corrected.

Line 75: Maybe '. . .strategies and analyses were conducted on both cruises. . ." rather than ". . .strategies and homogenous analyses were conducted . . .".
Corrected.

Line 85, 86: Maybe ". . . transects that employed a common sampling strategy of short and long duration stations." Rather than '. . .transects with similar sampling strategy of short and long duration stations."
Corrected.

Line 99: Given the very low nutrient concentrations it the reader would benefit from knowing the detection limits of the specific nutrient analyses employed.
The following line was added : "During BIOSOPE, nitrate (NO3-) detection limit was 0.05 µM (accuracy of ± 0.05 µM), phosphate (PO43-) detection limit was 0.02 µM (accuracy of ± 0.05 µM), orthosilicic acid (Si(OH)4) detection limit was 0.05 µM (accuracy of ± 0.05 µM). During OUTPACE the quantification limit was 0.05 µM for all nutrients."

Line 119: 'quarters' instead of '4'. 'Plastic' petri dishes right?
Corrected.

Line 127: What method was used to remove the interference from HF in the LSi colorimetric analysis: boric acid or dilution?
HF is diluted in filtered boric acid in our protocol. Added.

Line 131: Kinetic assays? Do you mean you conducted time courses to test the efficiency of different digestion times?
We meant that kinetic assays of the first NaOH extraction were carried out to determine on a few samples the optimal extraction time when all BSi is digested and prior to the linear increase of DSi showing the subsequent leaching of LSi. We have modified the sentence as follows :

"This is particularly the case when high LSi concentrations are present. Kinetic assays of orthosilicic acid were conducted in some samples from the Marquesas, Gyre, East-Gyre and near Upwelling stations during BIOSOPE to determine the optimal extraction time for BSi digestion, and results revealed negligible LSi interferences after an extraction time of 60 min."

Line 138-139: Please elaborate. It is unclear how the addition of Si was used to correct for dissolution in the face of the combined influence of dissolution of captured siliceous particles and the admixture of ambient water.
This section has been detailed as follows : "Biogenic silica export fluxes were determined from drifting sediment traps deployed for 4 consecutive days at three depths (153, 328, 519 m) at the three long duration stations of the OUTPACE cruise. For each trap samples, 160 mL were filtered onto 0.6 μm polycarbonate membranes and the filters were treated following a two-step digestion as described above. In addition to the BSi measurements, the dissolved Si measured directly in the supernatant of each trap at the time of subsampling minus the initial dissolved Si content in the seawater used to fill the trap was added to the final BSi concentrations, to account for BSi dissolution in the trap samples during storage. This step proved necessary, as BSi dissolution ranged between 16 and 90 % depending on the samples."

Line 141 Si & VSi rather than Si/VSi. Si/VSi looks like you are dividing one rate by the other. Line 150: 'averaged' instead of 'average'
Corrected.

Line 151. Many details are missing from this section of the methods. There is no indication of size fractions. Later in the paper it is claimed that kinetics were size fractioned like biomass, but I only see one set of kinetic curves and it is not clear what size fraction they represent (Fig 8.). Also in this section there is no mention of a 32Si addition.
We had mentioned that samples for kinetic curves were treated as described for in situ

samples (i.e. received a spike of 632 KBq and were filtered onto stacked 0.2, 2 and 10 µm filters). However due to experimental problems during filtration for the kinetic experiments, we have decided to remove the kinetic section alltogether (see below for further details).

Lines 187-196. The observation that the nitracline is much deeper than the silicicline is also observed in the Sargasso but not nearly to the same extent. It might be interesting to speculate as to why these depths differ in the discussion.
This difference is mostly the case during BIOSOPE, and could be the result of the strong Si-pump operating in the coastal upwelling, leading to advection of low-Si waters westwards towards the gyre (Dugdale and Wilkerson, 1998). However, the discussion part of this paper is dedicated to budgets, diatom community and evidence for Si uptake in the picosize fraction, hence bottom up control factors, linked to hydrology are not a key point of this paper. We feel it would be out of place to start this in the discussion.

Line 198: rather than 'The distribution of orthosilicic acid concentrations were less clearly contrasted, . . ." maybe 'Horizontal gradients were not as strong for orthosilicic acid. . ..".
Corrected.

Line 211: 'existed" rather than 'subsisted"
Corrected.

Line 212: "magnitude" instead of "amplitude".
Corrected.

Line 216: Maybe:" The Chla a distribution during BIOSCOPE was similar to that observed during OUPACE with extremely. . ."
Corrected.

Line 238: The units in the figures for BSi and LSi are in micromoles per liter whereas in the text the concentrations are discussed as nanomoles per liter. Be consistent. I would suggest changing the figure to nanomoles per liter as it gets rid of leading zeros.
The BSI/LSi figures have been changed to nmol L-1 to be consistent with text and the color bar increased in non linearity to better show low concentrations.

267: Maybe "LSi concentration was highest at both ends of the transect but concentrations remained below those of BSI with average LSi values. . ."
Corrected.

Line 271: Here the reader learns that kinetic experiments were size fractionated. Move this information to the Methods. More importantly only one size fraction is shown in fig 8. Where is the data from the other fraction? Also the legend for Fig 8 should indicate the fraction shown.

According to your comments and some other reviewer's comment on too high VSi values,

we have gone back to our raw data and found some inconsistencies in size-fractionated filtration between rSi and BSi. Some filters for rSi retained too much 32Si (either due to clogging or uncareful rinsing of samples), yielding too much rSi over BSi explaining the high VSi values. If the shape of the kinetic uptake is globally fine, we acknowledge this problem, but unfortunately see no way of correcting the data adequately. We have thus chosen to remove these data entirely.

Line 273; Maybe ". . . rank order of most productive stations follow the pattern observed for BSi with the highest values observed at UPW followed by UPX and MAR stations."
Corrected.

Line 281: It might be useful to readers if the specific rates are also translated into implied doubling times as this will give a sense of how fast or slow growth might be in the various areas.
Figure 8 (former kinetic figure) was replaced with the following showing K (doubling time) for each station.

[Figure]

Line 295: It is unclear what size fraction is shown in the Fig 8. Fix legend. Also where is the data for the other fraction. Please clarify.
It was the 0,2-2 µm size-fraction, but this figure has now been removed as explained above.

Line316: Maybe "the lowest" rather than"record low".
Corrected.

Line 359-360: Maybe "We obtained size-fractionated biomass and . . . OUTPACEprogram and size fractionated production. . .during the BIOSCOPEprogram.
Corrected.

Line362: This is a long paragraph. Maybe break here.
Corrected.

Line377: "documented" instead of "evidenced"
Corrected.

Line387: Here is a place where the influence of data from the Pacific on global budgets can be emphasized. The contribution fell when data from the NPSG was added and now it goes down again when the south Pacific is considered.
We have added this reference in the following sentence : " This budget has been recently revised down to 13 Tmol Si y-1 when considering budgets from the North Pacific (Nelson and Brzezinksi, 1997) reducing the contribution of subtropical gyres to 5-7% of global marine silica production (Brzezinksi et al., 2011; Tréguer and de La Rocha, 2013)."

Line390: The limited number of measurements is disappointing, but treated objectively.
We agree. Unfortunately the extent of funding at the time of the cruise and available quantity of 32Si did not allow for more sampling, nor replicate measurements.

Line 408: The flux is indeed incredibly low: wow! However, my appreciation of this is vague given that I do not understand the correction for dissolution in the traps discussed above.
This has been corrected as described above (answer to comment on lines 138-139).

Line 426: Maybe: DCM's are common in mid-ocean gyres and are known to be often dominated by pico-sized phytoplankton (Chavez et al. 1996), Studies documenting. . .. . ."
Corrected.

Line 448: As I read this discussion I find the text informative but I wonder if the stated trends might be reinforced through a non-dimensional scaling or other analysis that would provide an objective way to illustrate many of the inferred trends.

As we give mean and SD values for each zone, we feel that it is sufficient to characterize each region (that are defined hydrologically), as we are not trying to show any statistical differences between regions.

Line490: This is a very long paragraph. Maybe subdivide.
Corrected.

Line 542: Somewhere in this section the differences between the shape of the kinetic curves obtained herefor pico-size fraction and those for cultured Synechococcus should be discussed. In culture Synechococcus have linear uptake kinetics within the concentration range examined here whereas the data from the South Pacific clearly show a hyperbolic response. It's difficult to know for sure, but it might be possible that the organism responsible for Si uptake in the small size fraction in the South Pacific is something other than Synechococcus which would be very interesting.
These kinetics have been removed and can therefore not be mentionned in the discussion.

Line 545-546: Confusing sentence. Maybe "Significant BSi in the pico-sized fraction could be explained as an artifact from detritus or the contribution from a previously unrecognized taxa."

Corrected as follows: "The significant contribution of the pico-size fraction to the BSi stocks during both cruises could be explained by the presence of detrital components, however its contribution to Si(OH)4 uptake during BIOSOPE was really surprising but can be explained in the light of new findings"

Line 552 "by" rather than 'with"

Corrected.

Line 555: To finish this argument the expected shape of a curve resulting purely from fragments should be given. I would think the signal would then be very noisy and inconsistent which is not observed.

Corrected as follows: "If the former hypothesis was true, we would expect approximately the same amounts of broken fragments on all filters (i.e. for each increasing Si concentrations) and the shape of the curve would not be hyperbolic."

---

## Author Comment (AC2) · 9 Jul 2018

This is a good paper with timely and relevant information on a poorly studied region of the ocean. It has a both rate/biomass information along with floristic data, a combination not often available. It is unfortunate that the gyres had only a limited sampling.

The data is tantalizing in what is seen, but more sampling in this area is required to confirm the extremely low rates. It is a great deal of information to present and there are some areas where either the paper structure or text is confusing. As I note below, the methods need considerable improvement. The description of replication and error bars is unacceptably vague. Claims of differences are not justified by any statistical analysis. There are very few measures of variability given and reader is left to wonder if replicates were even collected. Each measurement should have a standard deviation, confidence interval or some other metric of variability. The methods need to explicitly note which samples were collected in duplicate, triplicate, etc. On line 150, the uptake measurements were noted to have a precision of 10-25% for the less productive station. What is it for more productive stations? To me, the use of separate figures for hydrology, nutrients, BSi, and rates creates difficulty in interpreting the information. Multiple pages of figures are needed to understand one cruise. It would be much more clear if all the data were in a single (or perhaps 2 adjacent) multi-panel figures. However, it requires a rewrite of the manuscript to discuss each cruise in parallel rather than dealing with hydrology, nutrients, etc from the two cruises together. Since the cruises are very separate in time and space, there is no reason to treat one data type at a time.

As the focus of this paper was to established basin/regional budgets for Si, we felt that discussing each parameter together, despite the temporal gap between the two cruises was more appropriate than presenting both cruises one after another, to get a sense of regional variability and that this would feel less repetitive for the reader.

Cell counts are very time consuming and tedious. Thus, it is always disappointing when the information is lumped into a single pool of diatoms in Figures.

We do have cell abundance per species, but first, the data set for the BIOSOPE cruise was already published once in Gomez et al . 2007 in a different form but with extensive tables of diatom taxa list (Table 2 therein). The data from OUTPACE is also available which is how we managed to estimate C biomass from each taxa and calculate relative contribution to POC in %. We felt that adding yet another graph with species relative contribution which requires a long legend and many colors to convey diatom diversity was not absolutely necessary within the scope of this paper, and that Fig.9 and Table 2 were sufficient. But since the data is available we have added a link for data access in the data availability statement.

From the methods, it is quite impossible to determine if diatom counts were from the same depths as the BSi or a subset.

Diatom cell counts were done on the same Niskins as BSi/LSi measurements. We have

added the following "Seawater samples *collected from the same CTDs and Niskins as particulate Si samples* were ..."

Please clarify this. If the data density is there, please add this to the figures as a contour plot.
Potential density isolines were added to Figures 3 and 4 as white contour overlays.

The data availability statement is not present nor is there an explanation of why it is not present. This is not acceptable and I cannot recommend publication until this condition is met (as noted in the Instructions to Authors for the journal).
This is an omission. The following statement has been added.
"Data is available upon request through both cruises databases. For the OUTPACE cruise, see   http://www.seanoe.org/data/00446/55743/
For the BIOSOPE cruise, see: http://www.seanoe.org/data/00446/55722/
These links will be running very shortly and before resubmission of revised version.

The figures lack panel labels except for Fig. 7. This needs to be corrected for publication.
This was added to all Figures and Figure legend.

Paragraph breaks need to be used for clarity, be they line spacing or indentations.
This has been corrected on several occasions where paragrahs were indeed very long.

Line: comment 19: Chlorophyll does not need to be capitalized.
corrected

33-34: I am not sure what "silica production. . .comparable to . . .all areas of diatomaceous sediment" means. One is a rate per volume per time, the other is mass per volume sediment. Please clarify.
Yes the sentence was not very clear. It is a statement from Nelson et al 95 that was not clearly rewritten, we have modified the sentence as follows :
" their silica production would be comparable to that of areas overlying major diatomaceous sediment accumulation zones."

39: need to define chl a abbreviation first.
Chla has now been written in full text, as definition is given later.

50-56: While these cited authors suggested these mechanisms may be leading to diatoms blooms, they have little direct experimental or observational data to this point. Wilson (2011) was later modified when a stratification value was discovered to be to high (see later work by Wilson et al. 2013) and Calil et al and Guidi et al. have done much more direct work on the role of mesoscale features than Krause et al. (2009, 2010). These are all key points to make,but please cite the correct papers.
We were actually trying to find references related to diatom blooms, even though we did not state this clearly in this sentence "Furthermore, oligotrophic regions are known to experience considerable variability in nutrient injections leading to episodical blooms depending on the

occurrence of internal waves (Wilson, 2011), meso-scale eddies (Krause et al., 2010) storms (Krause et al., 2009), or dust deposition events (Wilson, 2003)."
Looking at Guidi et al. 2012 (Does eddy-eddy interaction control surface phytoplankton distribution and carbon export in the North Pacific Subtropical Gyre?), we did not find any mention of the stimulation of a diatom but rather only mention of Trichodesmium bloom which was not the reference we were trying to highlight. We have added Calil et al 2011 in the list of references concerning local upwellings or dust deposition events.

104: please provide temperature and length of precombustion
4h at 250°C (now included)

116: cascading is probably not the best word choice. Sequential or stacked is more accurate.
"Cascading" has been replaced by the word "stacked"

123: digestion, not attack
corrected

122-134. I am curious how standards were treated to have the same pH value as the samples. Si is a pH sensitive assay, so this merits some consideration.
When filters are digested in NaOH, the supernatant is subsequentely neutralized with HCl so that pH is close to neutral. Tests of DSi standard curves made in the same NaOH-HCl matrix as samples wer not significantly different from standard curves done in milliQ water. However when running DSi analysis directly on seawater samples, then low silicate water is used as the standard curve matrix, as salinity does impact by ~10% the calculated DSi concentrations.

143: please specify how the light measurement was made and then applied to generate the incubation depth.
The text states : "Euphotic layer depths (Ze) were calculated as described in Raimbault et al. (2008) and Moutin et al. (2018)." We have added the following detail: "Sampling depths were adjusted to on deck incubators screen attenuation using measurements from an in situ PAR sensor (LI-COR instrument) mounted on the CTD frame."

151: Si uptake from the chlorophyll maximum. This description needs clarification. Was uptake measured as per section 3.7 or were changes in BSi measured as per section 3.6? The kinetic curve incubation lasted 8 hours, the in-situ incubation lasted dawn to dusk. Are there potential artifacts associated with the timing of division cycles? Later in the paper, it appears isotope uptake experiments were conducted, but the reader should not have to wait until then to know this. Finally, how relevant is this measurement to the waters above the DCM?
Si uptake kinetics were carried following the method described in setcion 3,6. For more clarity, several methodological additions were made in this section (32Si added, size of filters). The header for both section were clarified by mentionning that both bulk Si uptake and Si kinetic

experiments were size-fractionated. In situ incubations were combined with 14C, 13C, 15N and O2 fluxes measurements and immersed 24h, whereas kinetic experiments were carried out during 8 h in paralell in on deck incubators fitted with neutral nickel screens. Given the limited amount of available 32Si and the extremely low biomass observed during biosope except at the DCM, a choice was made to only perform size-fractionated kinetic experiments at the DCM level, or at both DCM and surface levels if ever some biomass was observed at the surface. About the incubation duration, we make the hypothesis that populations are not synchronous and that Si uptake should be constant over 24 h. Furthermore, upon addition of cold Si(OH)4, stimulation of uptake occurs and as thumb rule we try to stay below a threshold of 10% of consumption of initial available Si, which is why the incubations were stopped after 8h. We surmise that size fractionated kinetics uptake experiments carried out outside of the DCM would have yielded no measurable counts on the scintillation counter, given the absence of Si biomass, and expect our kinetic results to represent the most active diatom community present in the euphotic layer. However, the results of kinetic experiments were removed entirely due to experimental problems on borad (likely filtration issues) which yielded incoherent results between size-fractions and unrealistic VSi values.

162: please list the net specifics: mouth opening and mesh size
The following information was added : "During the OUTPACE cruise, additional WP2 phyto-net hauls (mouth opening 0.26 m$^2$ ; 35 µm mesh-size) were undertaken at each site integrating the 0-150 m water column"

216: This sentence is not clear. Please rephrase. It is apparently a comparison joined by the word than. I'm not sure what you are trying to say.
This sentence was rephrased as follows : "The Chl*a*  distribution during BIOSOPE was similar to that observed during OUPACE, with extremely low surface concentrations and a very deep Chl*a* maximum located between 180 - 200 m ranging between 0.15 and 0.18 µg L$^{-1}$."

266: attributed, not assimilated.
corrected

268: What is this unit of variability? Standard deviations? confidence interval? If you wish to say they are different, please refer to a statistical test showing this. The ± ranges overlap considerably. I am not convinced.
The unit is the standard deviation. Indeed, differences are not statistically significant, which is why we did not state "significantly" higher, but merely "higher". Yet we have corrected as follows : "The importance of the picoplanktonic Si biomass was higher in the SPG (36 ± 12 %, n=14) than over the MA (22 ± 10 %, n=5) but not statistically different (p > 0.05)."

269+: The same comment applies here. Are the duplicates? Triplicates? Error bars? Statistics? The rates have up to 25% precision errors, so this is important.
Unfortunately, given the limited amount of 32Si available, there were no duplicates on any of the 32Si uptake or kinetic measurements, which is why no error bar is indicated.

312+: contribution to biomass implies some conversion to a common currency (carbon, chl). How did you do this?

We have added the following explanation in section 3.9 : "Seawater samples were preserved with acidified Lugol's solution and stored at 4ºC. For the BIOSOPE cruise, a 500 mL aliquot of the sample was concentrated by sedimentation in glass cylinders for six days. Diatoms were counted following the method described by Gomez et al. (2007). For the OUTPACE cruise, a 100 mL aliquot of the sample was concentrated in an Utermöhl sedimentation chamber for 48h. Diatom sizes were measured for each species for an average number of 20 cells when possible, and converted to biovolume and C biomass following the method described in Leblanc et al. (2012). C biomass per species were then compared to chemically determined POC concentrations to yield a percent contribution to C biomass."

322: richness based on quantitative counts or the net tows? In either case, the authors need to specifiy the total number of cells examined. If it is 50 cells in one case and 500 in another, that will clearly influence the community richness observed.

By richness we only meant number of taxa as indicated in Figure 9c, and we did not use any common diversity indexes. The number of taxa was derived from quantitative cell counts from Niskin bottles sampled at the surface and DCM at each site.The richness as mentionned above is based solely on the number of different taxa present in an entire sedimentation cuve. For the BIOSOPE cruise, the counting method was to sediment 500 ml. Hence counted cells would be half of the cellular abundances indicated for this cruise. For the OUTPACE cruise, we counted 100 mL aliquots, hence counted cells were equaled to 1/10th of the cellular concentration. Methods were different as the data was obtained from the publically available cruise database and counting not performed in our research group. The details about the volume of the aliquots counted has now been added to section 3.9.

326: Dominance within the diatom community needs to be specified as based on abundance or size/surface area. One large Coscinodiscus or Rhizosolenia will equal many small bicapitate Nitzschia.

In this paragraph we indeed meant numerical abundance, which has been underscored now in the text. We did not present biomass estimates per species as the paper already presents and extensive dataset and only present global contribution to POC estimates. Indeed abundances were extremely low at some sites and subsequent conversions would be based on too few counts. It is agreed that large diatoms represent a disproportionate contribution to biomass compared to smaller species, but we did not feel that adding this level of detail in the discussion would be useful in the present paper. Furthermore this level of detail (specific contribution to biomass) is not available for the BIOSOPE cruise during which diatom sizes were not measured.

The Table 1 citation seems out of place. I think you mean Table 2.
Yes this has been corrected.

489: The authors may wish to consider the work of Shipe et al. (1999) where they noted large

rare diatoms contributed up to 26% of the Si uptake in the north Pacific. There is no information on these giant diatoms, either solitary or aggregated, from the south Pacific. Any observations they have on this would be very relevant.

Line 523 :    pennates were the most numerically abundant species (but also the most actively silicifying as evidenced by PDMPO staining - data not shown) but it is true that large sized diatoms such as Pseudosolenia calcar avis contribute disproportionately to C biomass compared to small pennates. In our calculation, 23 500 cells of Pseudo-nitzschia sp. represented 78% of total abundance but only 1 % of C biomass. Contrarily, 190 cells of Pseudo-solenia calcar-avis represented 0,6 % of total abondance but 90% of calculated biomass.

The following sentence was added line 576 : "However, it should be noted that if small fast growing pennates were numerically dominant, their relative contribution to C biomass was very small compared to that of few larger centrics such as Pseudosolenia calcar-avis, which when present dominated in terms of biomass, similarly to what had already been observed in the South Pacific with large Rhizosolenia (Shipe et al., 1999)."

593: This study is not a time series as per HOT and BATS, so the topic sentence implication that this work adds to time-series work in the south Pacific is not correct.

We did not imply that our work was similar to time-series data, and stated that we provide complementary data from two cruises, our point was to say that we propose comprehensive data of stocks, fluxes, kietincs and diatom taxonomy and abundance that are not usually provided all together in biogeochemical studies. We removed the term "complementary" from the sentence in order to remove the meaning that our data are similar to time-series.

Figures Fig. 4: The change in color scale is a bit confusing since the tendency to com-
pare the two transects. If Fig. 4 Outpace were the same color scale as the Bioscope
figure, then all the detail of the DCM would disappear. Likewise, the use of the Outpace
color scale for the Bioscope would create detail.

Indeed but the ranges are very different, making it difficult to convey the profile details with the same colors. We have tweeked the median and non-linear settings of the color bar of the BIOSOPE cruise, so that the green color is close to 0.2 µg $L^{-1}$ in both pannels.

Fig. 9: there are typos in the 2nd panel figure axis.

Corrected

---

## Author Comment (AC3) · 9 Jul 2018

Overall this is a solid study which presents a wealth of data from a vast and undersampled region. While not groundbreaking, it could be impactful if it spurs more study of Si cycling in this region. Generally, I agree with most of the study (the authors have done a commendable job with the cell count and taxonomy components) but have a few main comments:

- The contribution of Synechococcus: the authors have compelling data which is consistent with recent studies but this facet is under developed. Given the Silicon per cell for Synechococcus in the two publications these authors cite (Baines et al. 2012, Ohnemus et al. 2016), could they do a similar budget of Synechoccocus silica here? Given the size of this project, surely there must be some flow cytometry data.

Ohnemus 2016 : "However, Si contents were highly variable and generally uncorrelated with measured environmental variables, suggesting that less direct effects such as community structure may drive Si accumulation in these ecosystems." From the Table 2 in this paper, the authors estimate a contribution of Syn to the small fraction of BSi to range from 0.3 to 34 % though these contributions were always < 4% of total bSi, but they had Si per cell estimates from SXRF for each sample, and this ranged from 14 to 64 amol cell-1. We did calculate those estimates while writing this apper, but we felt that due to the strong variability and the absence of direct estimates for Si cellular quotas in Syn, this was not worth adding.
We however have these estimates and thus propose to add this sentence to the discussion as follows :
" Using the range of measured Si cellular content per *Synechococcus* cells given in Ohnemus et al. (2016) of 14 to 64 amol Si cell$^{-1}$ and *Synechocococcus* abundance data from the same casts obtained in flow cytometry (data courtesy of S. Duhamel, Lamont Doherty, NY), this yields a potential contribution of 3 to 14 % of *Synechococcus* to the small BSi fraction, which is close to the previous estimates."

- Additionally, the isotope data is excellent to have but the rates for the kinetic data are unrealistic and not adequately discussed. For instance, 3.0 d-1 implies 4.3 doublings per day, 4.0 d-1 implies 5.8 doublings per day. Among all the experiments shown in Fig 8, all rates are exceptionally high as to be not believable. I think the authors need to better justify whether these data are useful and, if not, then perhaps consider eliminating.

According to your comments on too high VSi values, we have gone back to our raw data and found some inconsistencies in size-fractionated filtration between rSi and BSi. Some filters for rSi retained too much 32Si (either due to clogging or uncareful rinsing of

samples), yielding too much rSi over BSi explaining the high VSi values. If the shape of the kinetic uptake is globally fine, we acknowledge this problem, but unfortunately see no way of correcting the data adequately. We have thus chosen to remove this data entirely. We have left the vertical profiles for rSi data and replaced figure 8 (kinetics) by a figure of the mean k (doubling time) for each station.

[Figure]

Beyond these issues, I have numerous minor comments:

Line 64-66: Baines's estimates were indirect and extrapolated significantly, and were based on bSi associated with living cells (instead of total bSi).

Noted, we have removed this citation in this sentence.

Line 118: why the difference in filter sizes? Does this affect your results and interpretations?

During the BIOSOPE cruise, filter size (0.2 and 2µm) were chosen to reflect the standard operational size-classes of pico- and nanophytoplankton. The published work in between those two cruises dealing with the presence of Si in the picosize fraction was published using filter sizes of 0.4 and 3 µm (Baines et al., 2012), and since we wanted to confirm or not these first results, we decided to use the same filter sizes for comparison.

Line 128: given such low bSi measured, it seems like this precision is quite high (i.e. high noise to signal ratio). May the authors please explain why they would not consider this an issue?

Going back to our data, we have corrected this statement as follows :"The detection limit was 1 nmol $L^{-1}$ for both BSi and LSi and quantification limits were 5 and 6 nmol $L^{-1}$ for BSi and LSi respectively." We are indeed very close to detection limits at some depths during both cruises, but we did not have replicates to estimate accuracy. We do feel that the accuracy estimates, that has been estimated to 4 and 6 nmol $L^{-1}$ respectively but for other cruises, if applied here would not significantly change any of the calculated budgets or

interpretation data, which would remain some of the lowest ever measured. We do observe that our measured in situ BSi concentrations were lowest than in most other papers, but we propose to add the following statement in the budget section, in order to underline the potential uncertainty on these baseline values. "

"For oceanic HNLC areas, values obtained (0.8 to 5.6 mmol Si $m^{-2}$ $d^{-1}$) cover the range of rates measured in HNLC to mesotrophic systems of the North Atlantic, Central Equatorial Pacific and Mediterranean Sea. However, integrated rates obtained for the oligotrophic area of the South Eastern Pacific Gyre are to our knowledge among the lowest ever measured, *even taking into account the error associated to budget estimates this close to analytical detection limits.*"

Line 149: Cerenkov counting is much less efficient than standard liquid scintillation methods correct? Given the low biomass (and thus low sample signal), did the Cerenkov background counts allow adequate resolution of analytically significant signals?

Cerenkov counting efficiency was estimated to be 42 % for this cruise, it is usually considered close to that value (~50%). Going over to liquid scintillation may have increased all cpm counts, but then also those of blanks, thus not improving the precision of the method.

Line 154: why go up to 36 uM? Are there prior studies which have gone this high? Recent work (Shrestha & Hildebrand 2015) show that above 25 uM diatoms start turning off silicon transporters.

Indeed the highest chosen concentration was probably too high, but it sometimes allow to show for a more linear response of Si uptake. In any case, the BIOSOPE cruise on which kientic experiments were made was conducted in 2005 so quite some time before the study you mention.

Line 229: given the high values, would the median (instead of average) be better here?

The median is now indicated at the end of the sentence, but is not notably different (13 instead of 17 nmol $L^{-1}$).

Line 275: 15 nmol/L/d given such low bSi means these cells are pretty active (e.g. 1 doubling per day)

Line 281, 298-299: Vmax is so high, it seems to be an error (see general comment).
See response above concerning VSi estimates

Line 296: it doesn't say in the figure caption that these are just for pico sizes, please Clarify.
This has been added to the Figure legend.

Line 353: what is the percent dissolution among these samples, could those be used to infer dissolution rates in the water column and compare to biomass-specific rates?

The % dissolution is indicated in the method section (line X) and was comprised between 16 and 90%. However it does not reflect in situ dissolution rates, but dissolution in trap samples kept at 4°C between sampling and analysis, hence not comparable to in situ rates.

Line 564: may you cite evidence for siliceous parmales in this region, aren't these only routinely observed in the subarctic North Pacific.
No they are not, there is evidence for large scale distribution, notably also for the Southern Ocean, but also large abundances have been oserved in the South Eastern Pacific (see Fig. 4 in Ichinomiya et al, 2016, ISME journal), where they can represent more than 1 % of total photosynthetic reads at both the surface and DCM depths.

Line 582: how so? There are two problems: the quotas published by Ohnemus et al. 2016 are low and the standing stock of picoplankton isn't high enough to consistently drawdown Si. Second, if these standing stocks did get high enough, then to remove Si, this material would need to be exported; yet the export rates quantified in this region were the lowest observed. This feels like a disconnect.

We agree that this role is probably not major, and have thus removed the term in the sentence. However, drawdown may be high by Synchococcus while it is also likely grazed and recycled in the surface layer. High temperatures are likely to remineralize a large part of assimilated Si in the surface layer, even though a previous drawdown and export prior to our study is necessary to explain the low silicic acid concentrations observed at the surface, and is not attributable to Synechococcus activity.

Figure 2, 3: could the color scale be more logarithmic (like in Figure 4) and similar to allow easier comparison?

We prefered giving the maximum details with color range for each graph, but have homogenized since all co-authors requested this.

Figure 7: perhaps a log scale to see the low values easier?
We modified the graph accordingly (see below)

[Figure]

Figure 9: please detail how the lower panel values were calculated

This is now described in the method section as follows :

"Diatom sizes were measured for each species for an average number of 20 cells when possible, and converted to biovolume and C biomass following the method described in Leblanc et al. (2012). C biomass per species were then compared to chemically determined POC concentrations to yield a percent contribution to C biomass."

---

## Author Comment (AC4) · 11 Jul 2018

Dear reviewers, please find a revised version with track changes and in pdf attached. K. Leblanc

Please also note the supplement to this comment: https://www.biogeosciences-discuss.net/bg-2018-149/bg-2018-149-AC4-supplement.zip